# BiCoLoR: Communication-Efficient Optimization with Bidirectional Compression and Local Training

## Abstract

Slow and costly communication is often the main bottleneck in distributed optimization, especially in federated learning where it occurs over wireless networks. We introduce BiCoLoR, a communication-efficient optimization algorithm that combines two widely used and effective strategies: local training, which increases computation between communication rounds, and compression, which encodes high-dimensional vectors into short bitstreams. While these mechanisms have been combined before, compression has typically been applied only to uplink (client-to-server) communication, leaving the downlink (server-to-client) side unaddressed. In practice, however, both directions are costly. We propose BiCoLoR, the first algorithm to combine local training with bidirectional compression using arbitrary unbiased compressors. this joint design achieves accelerated complexity guarantees in both convex and strongly convex heterogeneous settings. Empirically, BiCoLoR outperforms existing algorithms and establishes a new standard in communication efficiency.

## 1 Introduction

Distributed computing has become pervasive across scientific disciplines. A prominent example is Federated Learning (FL), which enables collaborative training of machine learning models in a distributed manner (Konečný et al., 2016a;b; McMahan et al., 2017; Bonawitz et al., 2017). This rapidly growing field leverages data residing on remote devices, such as smartphones or hospital workstations. While FL must address challenges such as preserving privacy and resisting adversarial threats, **communication efficiency** is arguably the most critical issue (Kairouz et al., 2021; Li et al., 2020a; Wang et al., 2021). Unlike centralized data-center setups, FL relies on parallel computation across client devices that must regularly exchange information with a distant central server. Communications typically occurs over bandwidth-limited and potentially unreliable networks, such as the internet or mobile networks. Consequently, communication remains the main bottleneck that hinders the widespread adoption of FL in large-scale consumer applications.

To mitigate communication overhead, two main strategies have gained prominence: (1) **Local Training (LT)**, which reduces communication frequency by allowing multiple (stochastic) gradient steps to be performed locally before transmitting updates; and (2) **Communication Compression (CC)**, where updates are transmitted in a compressed form rather than as full-dimensional vectors. A review of the literature on LT and CC is presented in Section 2.

It is important to distinguish between **uplink communication (UpCom)** (clients to server) and **downlink communication (DownCom)** (server to clients). UpCom tends to be slower, as clients must upload distinct messages to the server that needs to decompress each of them, whereas in DownCom, all clients typically receive the same message simultaneously. Several factors can reinforce this asymmetry, such as limitations in cache size and aggregation capabilities at the server, as well as differences in network protocols or service provider configurations across internet or cellular systems. Many methods have been proposed that use compression for UpCom only, considering that DownCom is cheap and can be neglected. This is of course not realistic, and in practical settings both ways are costly. That is why we focus in this work on **Bidirectional Compression (BiCC)**, applied to both downlink and uplink communication.

We measure the **total communication (TotalCom)** cost, in number of bits, as a weighted sum of the UpCom and DownCom costs:

$$\text{TotalCom} = \text{UpCom} + \alpha.\text{DownCom}, \tag{1}$$

for some parameter $\alpha \geq 0$, typically in $(0, 1]$. The case $\alpha = 1$ corresponds to the symmetric regime in which UpCom and DownCom are equally costly, while $\alpha = 0$ corresponds to ignoring DownCom completely, which is not realistic. We focus in this work in settings where $\alpha$ is not negligible, so that downlink compression is required. The model (1) has been proposed in Condat et al. (2022a; 2023), but counting the number of reals, not bits. Since real numbers are represented with finite precision (typically 32-bit floats), and quantization cannot reduce this size beyond a constant factor, see below, both measures are proportional. Nevertheless, counting bits provides a more accurate measure.

## 1.1 UNBIASED COMPRESSION

A common approach to reducing communication complexity in distributed learning is the use of lossy compression. This involves applying a (potentially randomized) compression operator $\mathcal{C} : \mathbb{R}^d \to \mathbb{R}^d$ to a $d$-dimensional vector $x$, such that transmitting $\mathcal{C}(x)$, encoded as a short bit stream, is significantly more efficient than sending the full vector $x$ (note that $\mathcal{C}(x)$ itself is the vector after decoding back to $\mathbb{R}^d$). Some compressors are unbiased; that is, $\mathbb{E}[\mathcal{C}(x)] = x$, where $\mathbb{E}[\cdot]$ denotes expectation. Others are biased (Beznosikov et al., 2020). A widely used sparsifying compressor is rand-$k$, where $k \in [d] := \{1, \ldots, d\}$. It randomly selects $k$ coordinates of $x$, scales them by $d/k$, and sets the remaining elements to zero. When the receiver knows which indices were selected—e.g., via a shared pseudo-random generator or by encoding them using a small overhead of at most $k \log_2 d$ bits—only those $k$ values need to be communicated, achieving a compression ratio of $d/k$. Aside sparsification, that reduces the number of reals, quantization is another widely-used technique that reduces the number of bits needed to represent these reals. For instance, Natural compression rounds a real number to a power of 2 probabilistically and represents it using 9 bits, instead of 32 bits for a full-precision float (Horváth et al., 2022).

To use compressors in iterative algorithms, we need to characterize them. For this, we define, for every $\omega \geq 0$, the set $\mathbb{U}(\omega)$ of **unbiased compressors** $\mathcal{C} : \mathbb{R}^d \to \mathbb{R}^d$ with bounded **relative variance** $\omega$; that is, such that $\mathbb{E}\left[\|\mathcal{C}(x) - x\|^2\right] \leq \omega \|x\|^2$, for every $x \in \mathbb{R}^d$. Many practical compressors belong to this class (Beznosikov et al., 2020; Albasyoni et al., 2020; Horváth et al., 2022; Condat et al., 2022b). Notably, rand-$k \in \mathbb{U}(\frac{d}{k} - 1)$, and Natural compression belongs to $\mathbb{U}(\frac{1}{8})$, so that its compression factor 32/9 is almost free. Composing two compressors in $\mathbb{U}(\omega_1)$ and $\mathbb{U}(\omega_2)$ yields a compressor in $\mathbb{U}(\omega)$ with $1 + \omega = (1 + \omega_1)(1 + \omega_2)$ (Condat & Richtárik, 2022). For instance rand-1 + Natural compression compresses a vector into $9(+\log_2 d)$ bits, with $\omega = \frac{9d}{8} - 1$. Importantly, there are lower bounds on the achievable compression level, and to compress vectors of $\mathbb{R}^d$ into $b$ bits, we have $\omega^{-1} \leq 4^{b/d} - 1$, so that $b(1 + \omega) = \Omega(d)$ (Safaryan et al., 2022; Albasyoni et al., 2020) (He et al., 2023b, Proposition 1).

Moreover, given a collection $(\mathcal{C}_i)_{i=1}^n$ of compression operators in $\mathbb{U}(\omega)$ for $\omega \geq 0$, to characterize the relative variance after averaging their outputs, we introduce the constant $\omega_{\text{av}} \geq 0$ such that

$$\mathbb{E}\left[\left\|\frac{1}{n}\sum_{i=1}^n \left(\mathcal{C}_i(x_i) - x_i\right)\right\|^2\right] \leq \frac{\omega_{\text{av}}}{n}\sum_{i=1}^n \|x_i\|^2, \tag{2}$$

for every $(x_i)_{i=1}^n \in \left(\mathbb{R}^d\right)^n$. This is not an additional assumption, because (2) is satisfied with $\omega_{\text{av}} = \omega$. But $\omega_{\text{av}}$ can be much smaller than $\omega$. In particular, if the $\mathcal{C}_i$ are mutually independent, (2) is satisfied with $\omega_{\text{av}} = \frac{\omega}{n}$.

We introduce BiCoLoR, a novel randomized algorithm designed for communication-efficient distributed optimization. It integrates LT and BiCC with arbitrary compressors in $\mathbb{U}(\omega)$ for UpCom and $\mathbb{U}(\omega_s)$ for DownCom, for any $\omega, \omega_s \geq 0$. Variance reduction with dual variables (Hanzely & Richtárik, 2019; Gorbunov et al., 2020; Gower et al., 2020) ensures convergence to the exact solution.

## 1.2 Problem formulation

We consider the server-client model in which $n \geq 1$ clients do computations in parallel and communicate in both directions with a server. We study distributed optimization problems of the form

$$\min_{x \in \mathbb{R}^d} \frac{1}{n} \sum_{i=1}^{n} f_i(x) + 2f_s(x) + g(x), \tag{3}$$

where $d \geq 1$ is the model dimension, $f_i : \mathbb{R}^d \to \mathbb{R}$ is the private function of client $i \in [n] \coloneqq \{1, \ldots, n\}$, $f_s$ is the private function of the server, $g$ is a public function that is known by the server and all clients. We suppose that a solution $x^\star$ of (3) exists and we make the following assumptions.

- All functions $f_i$, $f_s$, $g$ are convex and $L$-smooth, for some $L > 0$ ($\phi$ is $L$-smooth if $\nabla \phi$ is $L$-Lipschitz continuous: for every $x, y$, $\|\nabla \phi(x) - \nabla \phi(y)\| \leq L\|x - y\|$. The norm is the Euclidean norm throughout the paper).

- $f_i$, $f_s$, $g$ are $\mu$-strongly convex, for some $\mu \geq 0$ ($\phi$ is $\mu$-strongly convex if $\phi - \frac{\mu}{2}\|\cdot\|^2$ is convex). $\mu$ can be zero is the general convex case. If $\mu > 0$, the solution $x^\star$ of (3) exists and is unique, and we define the condition number $\kappa \coloneqq \frac{L}{\mu} \geq 1$.

The problem (3) captures a lot of important problems in machine learning, including empirical risk minimization (Sra et al., 2011; Shalev-Shwartz & Ben-David, 2014), and in many other fields. We aim at solving (3) efficiently in terms of communication, in the general **heterogeneous** setting; that is, the functions $f_i$, $f_s$, $g$ can be *arbitrarily different*, without any similarity assumption. We prove accelerated communication complexity of BiCoLoR in both strongly convex and general convex cases.

Thus, in the formulation (3), the server is an additional $(n+1)$-th machine with its own loss function $f_s$, connected to the $n$ clients in a star-network. The idea of using an auxiliary dataset at the server representative of the global data distribution has been considered in several works, especially to correct for discrepancies induced by partial participation (Zhao et al., 2018; Yang et al., 2021; 2024). This is different from our setting, where $f_s$ can be very different from the $f_i$. The function $g$ is shared by all machines, so that Client $i$ makes calls to $\nabla f_i$ and $\nabla g$, and the server makes calls to $\nabla f_s$ and $\nabla g$. $g$ can also be the loss with respect to a small auxiliary dataset shared by all machines, but this is not necessarily the case. *We stress that in the general convex case, $f_s$ and $g$ can be zero.* In the strongly convex case, one can choose $f_s = g = \frac{\mu}{2}\|\cdot\|^2$. So, the template problem (3) is versatile and includes the minimization of $\sum_{i=1}^{n} f_i$ as a particular case. The reason why $g$ is introduced in (3) is that an estimate $y$ of a solution $x^\star$ is computed by all machines, and every client compresses the difference between its local model estimate and $y$. These differences tend to zero, which is key to obtaining a variance-reduced algorithm converging to a solution $x^\star$ exactly.

## 1.3 Challenge and contribution

This work addresses the following question: *Can we combine LT and BiCC into a method that allows arbitrary unbiased compressors in UpCom and DownCom, provably benefits from the two techniques by exhibiting a state-of-the-art (SOTA) accelerated TotalCom complexity with nontrivial compression factors, and outperforms existing methods in practice?*

Our new algorithm BiCoLoR is the first to answer this complex question in the affirmative. This achievement overcomes significant challenges. For instance, with a standard application of random compression, the downlink and uplink compression errors *multiply* each other, as discussed in Section 2.2. In BiCoLoR, they are decoupled and only add up.

## 2 Related work

We review existing methods in the strongly convex case ($\mu > 0$), as the study of the linear convergence rates provides important insights. We use the notation $\tilde{\mathcal{O}}(\cdot) = \mathcal{O}(\cdot \log \epsilon^{-1})$, where $\epsilon > 0$ is the desired accuracy. We refer to Tyurin & Richtárik (2023) for a discussion of the general convex case.

## 2.1 LOCAL TRAINING

Local Training (LT) is a straightforward yet highly effective strategy. Instead of performing just a single (stochastic) Gradient Descent (GD) step between communication rounds, clients execute multiple steps locally. The core intuition is that these additional steps allow clients to transmit more informative updates, thereby reducing the total number of communication rounds required to achieve a target accuracy. LT is a core component of the popular FedAvg algorithm (McMahan et al., 2017), which is at the root of the immense success of FL. The idea of reducing the communication frequency was initially just heuristic, but a great deal of empirical evidence has demonstrated its practical effectiveness. LT was analyzed first in the homogeneous data regime, or under restrictive assumptions such as bounded gradient diversity (Haddadpour & Mahdavi, 2019), then in the more realistic regime of heterogeneous data (Khaled et al., 2019; Stich, 2019; Khaled et al., 2020; Li et al., 2020b; Woodworth et al., 2020; Gorbunov et al., 2021; Glasgow et al., 2022). The more GD steps are made locally, the closer the local models get to the minimizers of the local functions $f_i$, which is not the desired behavior. This effect, called *client drift*, has been quantified (Malinovsky et al., 2020). The next class of methods, which includes Scaffold (Karimireddy et al., 2020), S-Local-GD (Gorbunov et al., 2021) and FedLin (Mitra et al., 2021), implemented variance reduction techniques to correct for client drift, so that a consensus is reached and every local model converges to the exact global solution. These methods are not accelerated, however.

More recently, a significant advancement was introduced by Mishchenko et al. (2022) with Scaffnew, the first LT algorithm that converges linearly with accelerated TotalCom complexity $\tilde{\mathcal{O}}(d\sqrt{\kappa})$. In Scaffnew, communication occurs randomly after each GD step with only a small probability $p$, resulting in an average of $1/p$ local steps between communication rounds. The optimal dependency $\sqrt{\kappa}$ (Scaman et al., 2019) is achieved when $p = 1/\sqrt{\kappa}$. Scaffnew has then been extended in several ways (Malinovsky et al., 2022; Maranjyan et al., 2022; Condat & Richtárik, 2023; Yi et al., 2025).

## 2.2 COMPRESSION

A landmark development in the area of distributed algorithms using CC is the variance-reduced algorithm DIANA proposed in 2019 (Mishchenko et al., 2024). It achieves linear convergence with uplink compressors in $\mathbb{U}(\omega)$ for any $\omega \geq 0$. Its iteration complexity, with communication at every iteration, is $\tilde{\mathcal{O}}\left(\left(1 + \frac{\omega}{n}\right)\kappa + \omega\right)$. So, with independent rand-1 compressors, its UpCom complexity is $\tilde{\mathcal{O}}\left(\left(1 + \frac{d}{n}\right)\kappa + d\right)$, which significantly improves over $\tilde{\mathcal{O}}(d\kappa)$ of standard GD when the number of clients $n$ is large. This has highlighted CC as an acceleration mechanism. DIANA has been extended in various directions, including support for stochastic gradients and partial participation (Horváth et al., 2022; Gorbunov et al., 2020; Condat & Richtárik, 2022). The algorithm ADIANA (Li et al., 2020c), based on Nesterov's accelerated GD, has iteration complexity $\tilde{\mathcal{O}}\left(\left(1 + \frac{\omega}{\sqrt{n}}\right)\sqrt{\kappa} + \omega\right)$ (He et al., 2023b). For methods using independent uplink compressors in $\mathbb{U}(\omega)$, the lower bound $\tilde{\Omega}\left(\left(1 + \frac{\omega}{\sqrt{n}}\right)\sqrt{\kappa} + \omega\right)$ on the number of communication rounds has been established, so ADIANA is optimal in this sense. This translates into the UpCom complexity $\tilde{\mathcal{O}}\left(\left(1 + \frac{d}{\sqrt{n}}\right)\sqrt{\kappa} + d\right)$. Recently, linearly convergent algorithms using biased compressors have been proposed, such as EF21 (Richtárik et al., 2021; Fatkhullin et al., 2021; Condat et al., 2022b), but the theoretical understanding of these methods remains less mature and their acceleration potential is not clear.

Bidirectional Compression (BiCC) is more complicated. A standard way to implement it is as follows: after UpCom, the server forms an updated model by aggregation of the received compressed vectors from the clients. Then it compresses this model before DownCom to all clients for the next round. However, by doing so, the degradations due to uplink and then downlink compression pile up. A bidirectional extension of DIANA has been proposed, as part of the MURANA framework (Condat & Richtárik, 2022), with downlink compressors in $\mathbb{U}(\omega_s)$ for any $\omega_s \geq 0$. Its iteration complexity is $\tilde{\mathcal{O}}\left(\left(1 + \frac{\omega}{n}\right)(1 + \omega_s)\kappa + \omega\right)$, as reported in Table 1. We see that the uplink and downlink variances get multiplied with the $\left(1 + \frac{\omega}{n}\right)(1 + \omega_s)$ dependence. Other algorithms, such as Artemis (Philippenko & Dieuleveut, 2020) and DORE (Liu et al., 2020) have been proposed, with the same complexity. MCM Philippenko & Dieuleveut (2021) has the slightly better complexity shown in Table 1. Recently, EF21-P+DIANA was proposed (Gruntkowska et al., 2023), extending DIANA to BiCC using error feedback at the server. Its complexity is $\tilde{\mathcal{O}}\left(\left(1 + \frac{\omega}{n} + \omega_s\right)\kappa + \omega\right)$, with better decoupled complexity depending on the sum $1 + \frac{\omega}{n} + \omega_s$ of the variances instead of their product.

Table 1: Methods using arbitrary compressors $\mathcal{C}_i$ in $\mathbb{U}(\omega)$ for uplink and compressors $\mathcal{C}_s$ in $\mathbb{U}(\omega_s)$ for downlink communication, for arbitrary $\omega \geq 0$ and $\omega_s \geq 0$. All compressors are independent. The $\tilde{\mathcal{O}}$ notation hides the log factors, in particular $\log(\epsilon^{-1})$, in $\mathcal{O}(\cdot)$.

[a] 2Direction requires communication of full non-compressed vectors with a small probability.
[b] The reported complexity holds if $K$ satisfies the following conditions. For MURANA: $K = \Omega\left(\frac{d}{n}\right)$. For MCM: $K = \Theta(d)$ (no compression). For BiCoLoR: $K = \Omega\left(\frac{d}{\sqrt{\kappa}}\right)$.

| method | number of communication rounds | TotalCom with rand-$K$ [b] |
|---|---|---|
| MURANA | $\tilde{\mathcal{O}}\left(\left(1 + \frac{\omega}{n}\right)(1 + \omega_s)\kappa + \omega\right)$ | $\tilde{\mathcal{O}}(d\kappa)$ |
| MCM | $\tilde{\mathcal{O}}\left(\left(1 + \frac{\omega}{n} + \omega_s^{3/2} + \frac{\sqrt{\omega}\omega_s}{\sqrt{n}}\right)\kappa\right)$ | $\tilde{\mathcal{O}}(d\kappa)$ |
| EF21-P+DIANA | $\tilde{\mathcal{O}}\left((1 + \frac{\omega}{n} + \omega_s)\kappa + \omega\right)$ | $\tilde{\mathcal{O}}(d\kappa)$ |
| 2Direction [a] | $\tilde{\mathcal{O}}\left(\sqrt{(1+\omega)(1 + \frac{\omega}{n} + \omega_s)\kappa} + \omega + \omega_s\right)$ | $\tilde{\mathcal{O}}(d\sqrt{\kappa})$ |
| BiCoLoR | $\tilde{\mathcal{O}}\left(\sqrt{(1+\omega+\omega_s)(1 + \frac{\omega}{n} + \omega_s)\kappa}\right.$ $\left. +(1+\omega+\omega_s)(1 + \frac{\omega}{n} + \omega_s)\right)$ | $\tilde{\mathcal{O}}(d\sqrt{\kappa})$ |

Lastly, 2Direction was introduced (Tyurin & Richtárik, 2023), combining acceleration from momentum and decoupled BiCC, achieving the SOTA complexity in number of rounds shown in Table 1. With appropriate compressors, this gives a TotalCom complexity of $\tilde{\mathcal{O}}(d\sqrt{\kappa})$, which is the same, so neither worse nor better, than using no compression. Also, we note that 2Direction communicates full non-compressed vectors with a small probability. Ideally, a method would only communicate compressed vectors. BiCoLoR has this property and achieves the SOTA complexity $\tilde{\mathcal{O}}(d\sqrt{\kappa})$ as well. When $\alpha = 0$ and no downlink compression is applied, neither 2Direction nor BiCoLoR reverts to a known algorithm with unidirectional CC, and their complexity is worse than ADIANA, see the discussion in Section A.

In a different area, BiCC has been considered in the nonconvex Bayesian setting where compression consists of sampling from distributions (Egger et al., 2025).

### 2.3 Combining LT and CC

It has proved difficult to combine LT and CC while keeping their benefits, namely acceleration from $\kappa$ to $\sqrt{\kappa}$ and an UpCom complexity with a better dependence on $d$ when $n$ is large. Early combinations, such as Qsparse-local-SGD (Basu et al., 2020) and FedPAQ (Reisizadeh et al., 2020) fail to converge linearly. FedCOMGATE (Haddadpour et al., 2021) converges linearly but in $\tilde{\mathcal{O}}(d\kappa)$. Random reshuffling, a technique that can be viewed as a kind of LT, has been paired with CC (Sadiev et al., 2022; Malinovsky & Richtárik, 2022). The effective LT mechanism of Scaffnew has been combined with CC in CompressedScaffnew, achieving the UpCom complexity $\tilde{\mathcal{O}}\left(\sqrt{d\kappa} + \frac{d\sqrt{\kappa}}{\sqrt{n}} + d\right)$ (Condat et al., 2022a). It exhibits double acceleration with the $\sqrt{d}\sqrt{\kappa}$ dependence when $n$ is large. However, CompressedScaffnew uses a specific linear compression technique based on random permutations of the coordinates. Recently, LoCoDL was introduced (Condat et al., 2025), successfully combining LT in the spirit of Scaffnew with CC using arbitrary uplink compressors in $\mathbb{U}(\omega)$. Its UpCom complexity matches that of CompressedScaffnew. ADIANA has an even better complexity, that goes down to $\tilde{\mathcal{O}}(\sqrt{\kappa} + d)$ when $n$ is very large. Nevertheless, LoCoDL consistently outperforms ADIANA in practice and can therefore be regarded as the SOTA in terms of UpCom efficiency.

To the best of our knowledge, BiCoLoR is the first algorithm to combine the LT mechanism of Scaffnew, which yields $\sqrt{\kappa}$ acceleration, with BiCC using arbitrary unbiased compressors. Similar to LoCoDL, BiCoLoR uses an additional function $g$ in the problem and a variable $y$ shared by all clients, with compression of the differences between the local variables $x_i$ and $y$. But it has notable differences: the server is an additional machine with its own function $f_s$, and during each communication round, the variables $x_i$ and $y$ are updated using information on $x_s$ received from the server, while $x_s$ itself is updated using information on the $x_i$ received from the clients. Crucially, these

---

**Algorithm 1** BiCoLoR

1: **input:** stepsizes $\gamma, \eta, \eta_y, \rho, \rho_y > 0$; sequence of probabilities $(p_t)_{t \geq 1}$ sparsification level $k \in [d]$; local initial estimates $x_1^0, \ldots, x_n^0, x_s^0, y^0 \in \mathbb{R}^d$, initial control variates $u_1^0, \ldots, u_n^0, u_s^0, u_y^0 \in \mathbb{R}^d$ such that $\frac{1}{n} \sum_{i=1}^n u_i^0 + 2u_s^0 + u_y^0 = 0$

2: **for** $t = 0, 1, \ldots$ **do**

3:    **for** $i = 1, \ldots, n, s$, at clients and server in parallel, **do**

4:       $\hat{x}_i^t := x_i^t - \gamma \nabla f_i(x_i^t) + \gamma u_i^t$

5:       $\hat{y}^t := y^t - \gamma \nabla g(y^t) + \gamma u_y^t$   // the clients and server maintain identical copies of $y^t, u_y^t$

6:    **end for**

7:    flip a coin $\theta^t \in \{0, 1\}$, with $\mathrm{Prob}(\theta^t = 1) = p_{t+1}$

8:    **if** $\theta^t = 1$ **then**

9:       pick a subset $\Omega^t \subset [d]$ of size $k$ uniformly at random

10:      **for** $i = 1, \ldots, n$, at clients in parallel **do**

11:        $c_i^t := \mathcal{C}_{i,\Omega^t}^t \left( \hat{x}_{i,\Omega^t}^t - \hat{y}_{\Omega^t}^t \right)$

12:        send $c_i^t$ to the server

13:        receive $c_s^t$ from the server

14:        $x_{i,\Omega^t}^{t+1} := (1 - \rho)\hat{x}_{i,\Omega^t}^t + \rho \left( c_s^t + \hat{y}_{\Omega^t}^t \right)$

15:        $x_{i,[d]\setminus\Omega^t}^{t+1} := \hat{x}_{i,[d]\setminus\Omega^t}^t$

16:        $y^{t+1} := \hat{y}^t + \rho_y c_s$

17:        $u_i^{t+1} := u_i^t - \frac{p_{t+1}k\eta}{d\gamma}(c_i^t - c_s^t)$

18:        $u_y^{t+1} := u_y^t + \frac{p_{t+1}k\eta_y}{d\gamma}c_s^t$

19:      **end for**

20:      at server, in parallel to steps 11–18:

21:        $c_s^t := \mathcal{C}_{s,\Omega^t}^t \left( \hat{x}_{s,\Omega^t}^t - \hat{y}_{\Omega^t}^t \right)$

22:        send $c_s^t$ to all clients

23:        receive $(c_i^t)_{i=1}^n$ from the clients and aggregate $\bar{c}^t := \frac{1}{n} \sum_{i=1}^n c_i^t$

24:        $x_{s,\Omega^t}^{t+1} := \left( 1 - \frac{\rho + \rho_y}{2} \right) \hat{x}_{s,\Omega^t}^t + \frac{\rho + \rho_y}{2} \hat{y}_{\Omega^t}^t + \frac{\rho}{2} \bar{c}^t$

25:        $x_{s,[d]\setminus\Omega^t}^{t+1} := \hat{x}_{s,[d]\setminus\Omega^t}^t$

26:        $y^{t+1} := \hat{y}^t + \rho_y c_s^t$

27:        $u_s^{t+1} := u_s^t + \frac{p_{t+1}k\eta}{2d\gamma} \bar{c}^t - \frac{p_{t+1}k(\eta_y + \eta)}{2d\gamma} c_s^t$

28:        $u_y^{t+1} := u_y^t + \frac{p_{t+1}k\eta_y}{d\gamma} c_s^t$

29:    **else**

30:       $x_i^{t+1} := \hat{x}_i^t \ \forall i \in [n], \ x_s^{t+1} := \hat{x}_s^t, \ y^{t+1} = \hat{y}^t$

31:       $u_i^{t+1} := u_i^t \ \forall i \in [n], \ u_s^{t+1} := u_s^t, \ u_y^{t+1} := u_y^t$

32:    **end if**

33: **end for**

---

two updates are decorrelated, which is the key to obtain a decoupled TotalCom complexity, akin to EF21-P+DIANA and 2Direction.

## 3   PROPOSED ALGORITHM BiCoLoR

The proposed stochastic primal–dual method BiCoLoR is shown as Algorithm 1. At iteration $t$, Client $i$ computes $\hat{x}_i^t$ by a GD step on its individual function $f_i$, corrected by a dual variable $u_i^t$ that learns $\nabla f_i(x^\star)$. It also computes $\hat{y}^t$ by a GD step on $g$. The server, as a $(n + 1)$-th client, does the same using $f_s$ and $g$. Communication is triggered randomly with a small probability $p$. When it occurs, Client $i$ compresses $\hat{x}_i^t - \hat{y}^t$ and sends this compressed difference to the server, which aggregates their average $\bar{c}^t$. Unlike in many algorithms, $\bar{c}^t$ is not sent back to the clients to update their local variables. Instead, it is only used by the server to update its local variables $x_s^t$ and $u_s^t$. This is the compressed difference $c_s^t = \mathcal{C}_s^t(\hat{x}_s^t - \hat{y}^t)$, sent by the server to all clients, which they use to update their variables and $y$. So, UpCom and DownCom are independent and can be performed in parallel. We make the following assumption.

**Assumption 3.1** (compressors in BiCoLoR). There exist $\omega, \omega_s \geq 0$ such that $\mathcal{C}_i^t \in \mathbb{U}(\omega)$ and $\mathcal{C}_s^t \in \mathbb{U}(\omega_s)$, for every $t \geq 0$, $i \in [n]$. The compressors $(\mathcal{C}_1^t, \ldots \mathcal{C}_n^t, \mathcal{C}_s^t)$ are independent from the $(\mathcal{C}_1^{t'}, \ldots \mathcal{C}_n^{t'}, \mathcal{C}_s^{t'})$ if $t \neq t'$. Also, $\mathcal{C}_s^t$ is independent from the $(\mathcal{C}_i^t)_{i=1}^n$ for every $t \geq 0$. The $(\mathcal{C}_i^t)_{i=1}^n$ need not be mutually independent, this is characterized by $\omega_{\mathrm{av}}$ in (2).

More technically, BiCoLoR works as follows. We rewrite the problem (3) as

$$\min_{\mathbf{x}=(x_1,\ldots,x_n,x_s,y)} \frac{1}{n}\sum_{i=1}^{n} f_i(x_i) + 2f_s(x_s) + g(y) \quad \text{s.t.} \quad \mathbf{Dx} = 0, \tag{4}$$

where $\mathbf{D}$ is a linear operator such that $\mathbf{Dx} = 0$ if and only if $x_1 = \cdots = x_n = x_s = y$. We refer to the Appendix for the vector notations and definitions. The key property of our design is that $\mathbf{D}$ is chosen in such a way that applying $\mathbf{D}$ and its adjoint $\mathbf{D}^*$ can be approximated using unbiased stochastic estimates, given that the clients receive the compressed vector $c_s$ from the server, and nothing else, and the server receives the compressed vectors $c_i$ from the clients. Also, $(u_1^t, \ldots u_n^t, u_s^t, u_y^t)$ has to remain in the range of $\mathbf{D}^*$, which means that $\frac{1}{n}\sum_{i=1}^{n} u_i^t + 2u_s^t + u_y^t = 0$, for every $t \geq 0$. This is why the idea that the server compresses the average of the $c_i^t$ and sends it back to the clients, instead of $c_s^t$, does not work, for instance. The operator norm of $\mathbf{D}$ is 2, implying that enabling BiCC incurs a twofold slowdown of BiCoLoR without compression, relative to vanilla GD. The iteration of BiCoLoR takes the form

$$\left|\begin{array}{l} \hat{\mathbf{x}}^t := \mathbf{x}^t - \gamma\nabla\mathbf{f}(\mathbf{x}^t) + \gamma\mathbf{D}^*\mathbf{u}^t \\ \text{flip a coin } \theta^t \in \{0,1\}, \text{ with } \mathrm{Prob}(\theta^t = 1) = p_{t+1} \\ \textbf{if } \theta^t = 1: \quad \mathbf{x}^{t+1} :\approx \hat{\mathbf{x}}^t - \rho\mathbf{D}^*\mathbf{D}\hat{\mathbf{x}}, \; \mathbf{u}^{t+1} :\approx \mathbf{u}^t - \frac{p\boldsymbol{\eta}}{\gamma}\mathbf{D}\hat{\mathbf{x}} \\ \textbf{else}: \quad \mathbf{x}^{t+1} := \hat{\mathbf{x}}^t, \; \mathbf{u}^{t+1} := \mathbf{u}^t \end{array}\right. ,$$

where $:\approx$ means that an unbiased stochastic estimate of the right-hand side, built from the compressed vectors, is used for the update; see the Appendix for precise definitions. The two stochastic estimates used for $\mathbf{x}$ and $\mathbf{u}$ are different. This is because an estimate of $\mathbf{D}^*\mathbf{D}\hat{\mathbf{x}}$ is needed to update $\mathbf{x}$, whereas an estimate of $\mathbf{D}\hat{\mathbf{x}}$ is needed to update $\mathbf{u}$, with $\mathbf{D}^*$ applied exactly to it. So, the estimate used for $\mathbf{x}$ is less noisy, with a variance that depends on $\omega_{\mathrm{av}} + \omega_s$ instead of $\omega + \omega_s$.

In BiCoLoR, when communication happens, we allow the selection of a subset $\Omega^t$ of size $k \in [d]$ of coordinates to be processed. The other coordinates are updated as if $\theta^t = 0$. For a vector $x \in \mathbb{R}^d$, its restriction $x_\Omega \in \mathbb{R}^d$ denotes $x$ with the coordinates not in $\Omega$ set to zero. Accordingly, $\mathcal{C}_\Omega(x_\Omega)$ applies compression only to the subset of coordinates in $\Omega$ and returns a $k$-sparse vector with coordinates not in $\Omega$ set to zero. This approach allows to sparsify communication, but the same $k$ random coordinates are used for all vectors, in UpCom and DownCom. By contrast, we assume in Assumption 3.1 that if rand-$k$ compressors are used instead, achieving the same sparsification factor, the uplink and downlink compressors are independent. The interest of sparsifying via $k$ and not the compressors is that variance reduction of the compression error can be bypassed, leading to larger stepsizes $\eta$ and $\rho$.

## 4 CONVERGENCE ANALYSIS AND COMPLEXITY OF BiCoLoR

### 4.1 ACCELERATED LINEAR CONVERGENCE IN THE STRONGLY CONVEX CASE

**Theorem 4.1** (linear convergence of BiCoLoR). *Suppose that $\mu > 0$ and let $x^\star$ be the unique solution to (3). In BiCoLoR, suppose that Assumption 3.1 holds, $0 < \gamma < \frac{2}{L}$, $p_t \equiv p \in (0,1]$ is constant, and*

$$\rho = \rho_y = \frac{1}{2 + \omega_{\mathrm{av}} + 2\omega_s}, \quad \eta = \eta_y = \frac{1}{(1 + 2\omega + 2\omega_s)(2 + \omega_{\mathrm{av}} + 2\omega_s)}. \tag{5}$$

*For every $t \geq 0$, define the Lyapunov function*

$$\Psi^t := \frac{1}{\gamma}\left(\sum_{i=1}^{n}\left\|x_i^t - x^\star\right\|^2 + 2n\left\|x_s^t - x^\star\right\|^2 + n\left\|y^t - x^\star\right\|^2\right) + \frac{d^2\gamma}{p^2 k^2\eta}\left(\sum_{i=1}^{n}\left\|u_i^t - u_i^\star\right\|^2 + n\left\|u_y^t - u_y^\star\right\|^2\right),$$

*where $u_y^\star := \nabla g(x^\star)$ and $u_i^\star := \nabla f_i(x^\star)$. Then BiCoLoR converges linearly: for every $t \geq 0$,*

$$\mathbb{E}\left[\Psi^t\right] \leq c^t\Psi^0, \quad \text{where} \quad c := \max\left((1-\gamma\mu)^2, (1-\gamma L)^2, 1 - \frac{p^2 k^2\eta}{d^2}\right) < 1. \tag{6}$$

*In addition, for every $i \in [n]$, $(x_i^t)_{t\in\mathbb{N}}$, $(x_s^t)_{t\in\mathbb{N}}$ and $(y^t)_{t\in\mathbb{N}}$ converge to $x^\star$, $(u_i^t)_{t\in\mathbb{N}}$ converges to $u_i^\star$, and $(u_y^t)_{t\in\mathbb{N}}$ converges to $u_y^\star$, almost surely.*

Thus, BiCoLoR has the same rate $\max(1 - \gamma\mu, \gamma L - 1)^2$ as vanilla GD, as long as $p^{-1}$ and the compression variances are below some threshold. The iteration complexity of BiCoLoR to reach $\epsilon$-accuracy, i.e. $\mathbb{E}[\Psi^t] \leq \epsilon$, with $\gamma = \Theta(\frac{1}{L})$, is

$$\mathcal{O}\left(\left(\kappa + \frac{d^2(1 + \omega + \omega_s)(1 + \omega_{\mathrm{av}} + \omega_s)}{p^2 k^2}\right) \log \frac{\Psi^0}{\epsilon}\right). \tag{7}$$

With $k = d$ and $\omega_s = 0$, this is the same complexity as LoCoDL. The complexity in number of communication rounds is $p$ times the iteration complexity, so the best value of $p$ balances the two terms in (7); that is, $p \propto \min\left(\frac{d\sqrt{(1+\omega+\omega_s)(1+\omega_{\mathrm{av}}+\omega_s)}}{k\sqrt{\kappa}}, 1\right)$. With this choice, the number of communication rounds is

$$\mathcal{O}\left(\left(\frac{d\sqrt{(1 + \omega + \omega_s)(1 + \omega_{\mathrm{av}} + \omega_s)}}{k}\sqrt{\kappa} + \frac{d^2(1 + \omega + \omega_s)(1 + \omega_{\mathrm{av}} + \omega_s)}{k^2}\right) \log \frac{\Psi^0}{\epsilon}\right). \tag{8}$$

Assuming that the vectors compressed by $\mathcal{C}_i$ and $\mathcal{C}_s$ are encoded into $b$ and $b_s$ bits, respectively, the **TotalCom** complexity, as defined in (1), is $(b + \alpha b_s)$ times the complexity in (8). As mentioned in Section 1.1, $b(1 + \omega) = \Omega(d)$, and the rand-$K$ compressor achieves this bound with $b = \Theta(K)$ (ignoring potential additional $K \log_2 d$ bits) and $1 + \omega = \frac{d}{K}$. So, let us look at the TotalCom complexity with respect to $K \in [d]$ and $K_s \in [d]$, assuming that the compressors $\mathcal{C}_i$ and $\mathcal{C}_s$ are independent rand-$K$ and rand-$K_s$, respectively, and $k = d$. In addition, we assume that $\alpha \leq 1$ and $K_s \geq K$. Then TotalCom is

$$\tilde{\mathcal{O}}\left((K + \alpha K_s)\left(\sqrt{\frac{d}{K}\left(\frac{d}{nK} + \frac{d}{K_s}\right)}\sqrt{\kappa} + \frac{d}{K}\left(\frac{d}{nK} + \frac{d}{K_s}\right)\right)\right).$$

If in addition $nK \geq K_s$, this simplifies to $\tilde{\mathcal{O}}\left((K + \alpha K_s)\left(\frac{d\sqrt{\kappa}}{\sqrt{KK_s}} + \frac{d^2}{KK_s}\right)\right)$. In the unlikely case $\alpha \leq \frac{1}{d}$, with $K_s = d$ (no downlink compression since DownCom is very cheap) and $K = 1$ (which implies $n \geq d$), the complexity becomes $\tilde{\mathcal{O}}\left(\sqrt{d}\sqrt{\kappa} + d\right)$, which shows double acceleration with respect to $d$ and $\kappa$. In particular, when $\alpha = 0$, this is the same complexity as CompressedScaffnew and LoCoDL (Condat et al., 2025). In the general case we are interested in, where $\alpha \in (0, 1]$ is not tiny, notably the important case $\alpha = 1$, we cannot expect acceleration with respect to $d$, see the discussion in Section A. So, we suggest the following values.

**Corollary 4.2.** *In the conditions of Theorem 4.1, suppose that $\alpha \in (0, 1]$, $k = d$, $\gamma = \frac{1}{L}$, the $\mathcal{C}_i^t$ and $\mathcal{C}_s^t$ are independent* rand-$K$ *and* rand-$K_s$ *compressors, respectively, with*

$$K_s = \left\lceil \frac{d}{\sqrt{\kappa}} \right\rceil, \quad K = \left\lceil \frac{\max\left(\alpha, \frac{1}{n}\right)d}{\sqrt{\kappa}} \right\rceil, \quad and \ \ p = \min\left(\frac{1}{\sqrt{\eta\kappa}}, 1\right). \tag{9}$$

*Then the analysis above applies, with $nK \geq K_s \geq K$ and $K \geq \alpha K_s$, and the TotalCom complexity of* BiCoLoR *is*

$$\mathcal{O}\left(\left(\frac{d\sqrt{K}\sqrt{\kappa}}{\sqrt{K_s}} + \frac{d^2}{K_s}\right) \log \frac{\Psi^0}{\epsilon}\right) = \mathcal{O}\left(d\sqrt{\kappa} \log \frac{\Psi^0}{\epsilon}\right). \tag{10}$$

An alternative is to make use of sparsification with the parameter $k$ in BiCoLoR:

**Corollary 4.3.** *In the conditions of Theorem 4.1, suppose that $\alpha \in (0, 1]$, $\gamma = \frac{1}{L}$, the compressors satisfy $\omega = \mathcal{O}(1)$, $\omega_s = \mathcal{O}(1)$ (e.g. quantization), $k = \left\lceil \frac{d}{\sqrt{\kappa}} \right\rceil$, $p = \min\left(\frac{d}{k\sqrt{\eta\kappa}}, 1\right)$. Then the TotalCom complexity of* BiCoLoR *is* $\mathcal{O}\left(d\sqrt{\kappa} \log \frac{\Psi^0}{\epsilon}\right)$.

We discuss the $\tilde{\mathcal{O}}(d\sqrt{\kappa})$ TotalCom complexity and explain why it is unlikely to be improved in Section A.

## 4.2 ACCELERATED SUBLINEAR CONVERGENCE IN THE GENERAL CONVEX CASE

For every $x \in \mathbb{R}^d$, $x' \in \mathbb{R}^d$, we define the Bregman distance of a convex differentiable function $\phi$ as $\mathcal{D}_\phi(x, x') := \phi(x) - \phi(x') - \langle\nabla\phi(x'), x - x'\rangle \geq 0$. If $\phi$ is $L$-smooth, we have $\langle\nabla\phi(x) - \nabla\phi(x'), x - x'\rangle \geq \mathcal{D}_\phi(x, x') + \frac{1}{2L}\|\nabla\phi(x) - \nabla\phi(x')\|^2$.

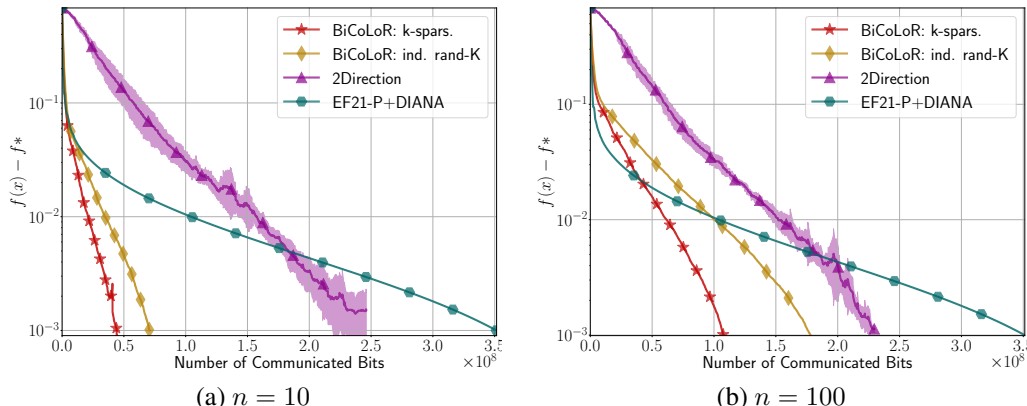

(a) $n = 10$  (b) $n = 100$

Figure 1: Logistic regression on the `real-sim` dataset. The compression scheme combines sparsification with $K = 1000$ and Natural Compression.

**Theorem 4.4** (accelerated sublinear convergence of BiCoLoR). *In BiCoLoR, suppose that Assumption 3.1 holds, $0 < \gamma \le \frac{1}{L}$, $k = d$,*

$$\rho = \rho_y = \frac{1}{2 + \omega_{\text{av}} + 2\omega_s}, \quad \eta = \eta_y = \frac{C}{(1 + 2\omega + 2\omega_s)(2 + \omega_{\text{av}} + 2\omega_s)}, \tag{11}$$

*for some constant $C \in (0, 1)$, and that for every $t \ge 1$,*

$$p_t = \sqrt{\frac{b}{a+t}} \in (0, 1],$$

*for some $b \ge \frac{1}{\eta}$ and $a \ge b - 1$. Let $x^\star$ be a solution of (3), and $u_y^\star := \nabla g(x^\star)$, $u_i^\star := \nabla f_i(x^\star)$ for every $i \in [n]$. Then BiCoLoR converges sublinearly: for every $\epsilon > 0$, by choosing $\tilde{t}$ uniformly at random in $\{0, \ldots, T-1\}$, where*

$$T := \left\lceil \frac{1}{2\epsilon} \left( \frac{1}{\gamma} \sum_{i=1}^n \|x_i^0 - x_i^\star\|^2 + \frac{2n}{\gamma} \|x_s^0 - x_s^\star\|^2 + \frac{n}{\gamma} \|y^0 - y^\star\|^2 + \frac{\gamma a}{\eta b} \sum_{i=1}^n \|u_i^0 - u_i^\star\|^2 + \frac{n\gamma a}{\eta b} \|u_y^0 - u_y^\star\|^2 \right) \right\rceil,$$

*we have*

$$\mathbb{E}\left[ \sum_{i=1}^n \mathcal{D}_{f_i}\left(x_i^{\tilde{t}}, x^\star\right) + 2n\mathcal{D}_{f_s}\left(x_s^{\tilde{t}}, x^\star\right) + n\mathcal{D}_g\left(y^{\tilde{t}}, x^\star\right) \right] \le \epsilon. \tag{12}$$

*and*

$$\mathbb{E}\left[ \frac{1}{\gamma} \sum_{i=1}^n \left\| x_i^{\tilde{t}} - x_s^{\tilde{t}} \right\|^2 \right] = \mathcal{O}\left( \sqrt{\epsilon} \right), \quad \mathbb{E}\left[ \frac{n}{\gamma} \left\| x_s^{\tilde{t}} - y^{\tilde{t}} \right\|^2 \right] = \mathcal{O}\left( \sqrt{\epsilon} \right) \tag{13}$$

*Moreover, the expectation of the number of communication rounds over the first $T \ge 1$ iterations is $\sum_{t=1}^T p_t = \Theta(\sqrt{T})$, so that (12) is achieved with $\Theta(\sqrt{T}) = \Theta\left( \frac{1}{\sqrt{\epsilon}} \right)$ communication rounds (we refer to the proof in the Appendix for the constants in $\mathcal{O}$). Moreover, if $\gamma = \Theta(\frac{1}{L})$, $x^0 := x_1^0 = \cdots = x_n^0 = x_s^0 = y^0$ and $u_1^0 = \nabla f_1(x^0), \ldots, u_n^0 = \nabla f_n(x^0)$, $u_y^0 = \nabla g(x^0)$, $\sqrt{T} = \Theta\left( \sqrt{\frac{L}{\epsilon}} \|x^0 - x^\star\| \right)$. This is the same accelerated complexity as* 2Direction *(Tyurin & Richtárik, 2023).*

Due to space constraints, **experiments** are presented in the Appendix. Some results are shown in Figure 1; see the Appendix for details and other results.

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

# Appendix

## CONTENTS

## A  DISCUSSION OF THE $\tilde{\mathcal{O}}(d\sqrt{\kappa})$ TOTALCOM COMPLEXITY

Let us comment on existing lower and upper bounds for the TotalCom complexity in the strongly convex setting.

As discussed in Section 2.2, For methods using uplink compression only with independent compressors $\mathcal{C}_i^t$ in $\mathbb{U}(\omega)$, the lower bound, achieved by ADIANA, for the number of communication rounds is $\tilde{\Omega}\big((1 + \frac{\omega}{\sqrt{n}})\sqrt{\kappa} + \omega\big)$. This translates into an UpCom (or equivalently TotalCom with $\alpha = 0$) complexity of $\tilde{\mathcal{O}}\big((1 + \frac{d}{\sqrt{n}})\sqrt{\kappa} + d\big)$. BiCoLoR, like 2Direction, achieves the same complexity $\tilde{\mathcal{O}}\big(\sqrt{d}\sqrt{\kappa} + d\big)$ as LoCoDL if $n \geq d$ and $\alpha = 0$. This is worse than ADIANA, but LoCoDL outperforms ADIANA in practice (Condat et al., 2025). In any case, there remains a theoretical gap between unidirectional and bidirectional CC.

Without downlink compression, or little compression with $\omega_s = \mathcal{O}(1)$ such as quantization solely, by choosing the $\mathcal{C}_i^t$ as independent rand-$K$ compressors for $K = \max(\frac{d}{n}, 1, d\alpha)$ (which means no uplink compression if $\alpha = 1$), BiCoLoR achieves the same TotalCom complexity $\tilde{\mathcal{O}}\big(\frac{d\sqrt{\kappa}}{\sqrt{n}} + \sqrt{d}\kappa + d + \sqrt{\alpha}d\sqrt{\kappa}\big)$ as CompressedScaffnew (Condat et al., 2022a). For not-so-small values of $\alpha$, this reverts to $\tilde{\mathcal{O}}(d\sqrt{\kappa})$. Moreover, this is obtained by essentially disabling compression and using LT only. This is not in the spirit of what we want to achieve, which is the best TotalCom complexity with nontrivial levels of compression in both ways. In Corollaries 4.2 and 4.3, BiCoLoR has complexity $\tilde{\mathcal{O}}(d\sqrt{\kappa})$ with large levels of compression, even when $\alpha = 1$. Still, there is a gap here too, between the regime $\alpha = 0$ where acceleration with respect to $d$ is possible thanks to compression, and $\alpha = 1$, where the TotalCom complexity is $\tilde{\mathcal{O}}(d\sqrt{\kappa})$ and compression is at best harmless. Nesterov's accelerated GD and Scaffnew (Mishchenko et al., 2022) have this complexity, and they don't use compression.

It has been shown that without assuming independence of the $\mathcal{C}_i^t$, a lower bound for the number of communication rounds is $\tilde{\Omega}\big((1+\omega)\sqrt{\kappa}\big)$, which gives an UpCom complexity of $\tilde{\Omega}(d\sqrt{\kappa})$ (He et al., 2023a). This complexity is achieved by Nesterov's accelerated GD and Scaffnew (Mishchenko et al., 2022), which do not use compression. This may indicate that many independent compressors run in parallel are required to hope for a decrease of the dependence with respect to $d$. We may consider the idea that the server sends different messages compressed with independent compressors, instead of using a single compressor $\mathcal{C}_s^t$. However, a negative result has been established, in the different nonconvex setting though: any method in which the server sends a compressed vector to each client, possibly obtained using different compressors in $\mathbb{U}(\omega_s)$, requires at least $\Omega\big((1+\omega_s)L/\epsilon\big)$ rounds to find a stationary point (Gruntkowska et al., 2024, Theorem 3.1); see also Huang et al. (2022). This suggests that there is little hope of improving the downlink dependence on $\omega_s$ as $n$ increases.

## B  EXPERIMENTS

### B.1  STRONGLY CONVEX CASE

We evaluate our proposed method, BiCoLoR, against 2Direction and EF21-P+DIANA on a regularized logistic regression problem of the form (3). The loss function of Client $i$ is

$$f_i(x) = \frac{1}{m}\sum_{s=1}^{m}\log\Big(1+\exp\big(-b_{i,j}a_{i,j}^\top x\big)\Big) + \frac{\mu}{2}\|x\|^2, \tag{14}$$

and we take $f_s = g = \frac{\mu}{2}\|x\|^2$. For the other algorithms, which do not use $f_s$ and $g$, we replace $\mu$ by $4\mu$ in the functions $f_i$, so that the problem solved by the different algorithms is exactly the same. In (14), $n$ is the number of clients, $m$ is the number of data points per client, $a_{i,j} \in \mathbb{R}^d$ and $b_{i,j} \in \{-1, +1\}$ are data samples, and $\mu$ is set so that the condition number is $\kappa = 4.10^6$.

Our experiments use datasets from the LibSVM library (Chang & Lin, 2011) (3-clause BSD license). Each dataset is first shuffled, then divided into $n$ pieces of same size $m$ assigned to the $n$ clients, discarding any leftover data points to ensure $m$ is an integer.

For BiCoLoR, we consider two compression strategies. We focus on the case $\alpha = 1$, so that the compression level is the same downlink and uplink.

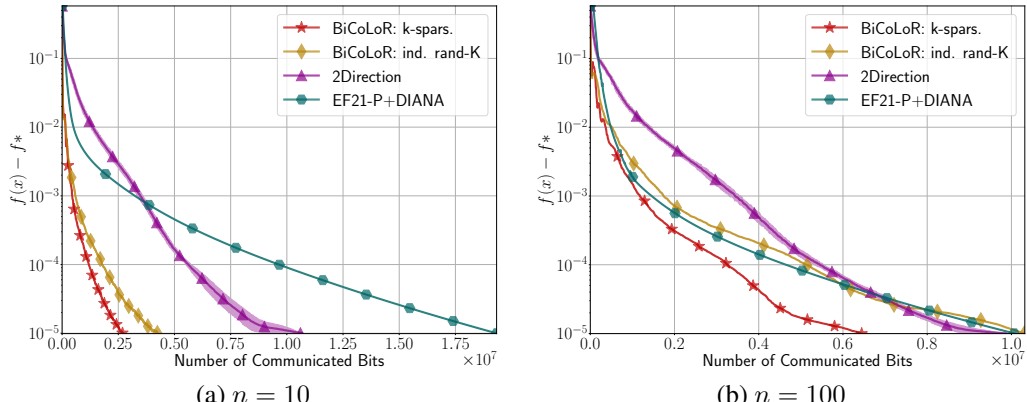

(a) $n = 10$                                                   (b) $n = 100$

Figure 2: Logistic regression on the `w8a` dataset. The compression scheme combines sparsification with $K = 100$ and Natural Compression.

1. Sparsification is performed using a low value of the parameter $k$, like in Corollary 4.3. The compressors are independent Natural compressors, performing quantization of reals on 9 bits, as explained in Section 1.1.

2. $k = d$ and the compressors are the composition of independent `rand-K` compression with same $K$, like in Corollary 4.2, and Natural compression. See in Section 1.1 for the variance of a composition.

For a given $K \in [d]$ and $k = K$ in the first strategy, the compression level is the same, but the difference is that every machine selects $K$ random coordinates independently in the second strategy, whereas the $K$ randomly selected coordinates are the same for all machines in the first one. Independence is beneficial but requires lower stepsizes $\eta$ and $\rho$, so we don't know a priori which of the two strategies is best.

For the other methods 2Direction and EF21-P+DIANA, we also use a combination of `rand-K` and Natural compression, both downlink and uplink (with appropriate scaling to make the downlink compressor contractive).

We tuned the stepsizes ($\gamma$ for BiCoLoR) for all algorithms. All other parameters are set to their best theoretical value. The algorithms are initialized with zero vectors.

Figures 1 and 2 show the results on the `real-sim` dataset (72,309 samples and $d = 20,958$ features) and `w8a` dataset (49,749 samples and $d = 300$ features), respectively.

BiCoLoR with the first strategy (using the parameter $k$ for sparsification instead of the compressors, red curves in the plots) outperforms the other algorithms. So, our theoretical findings are confirmed in practice, and BiCoLoR establishes a new state of the art for optimization with bidirectional compression.

### B.2 GENERAL CONVEX CASE

In this section, we aim to demonstrate the acceleration benefit of using a decreasing sequence $p_t$, as defined in Theorem 4.4, in the general convex setting. To this end, we conduct experiments using a convex loss function by removing the regularization term from the logistic regression objective (14), i.e., setting $\mu = 0$. The experiments are performed on the `w8a` dataset with $n = 10$.

We compare BiCoLoR with decreasing $p_t$ (as in Theorem 4.4) against BiCoLoR with constant $p_t \equiv p$, where $p$ is chosen so that both variants perform the same total number of local steps. For compression, like in Section B.1, we consider sparsification with the parameter $k$ and Natural compression in BiCoLoR, and the other algorithms use compositions of `rand-K` and Natural compression. We tuned the stepsizes ($\gamma$ for BiCoLoR) for all methods, and all other parameters are set to their best theoretical values.

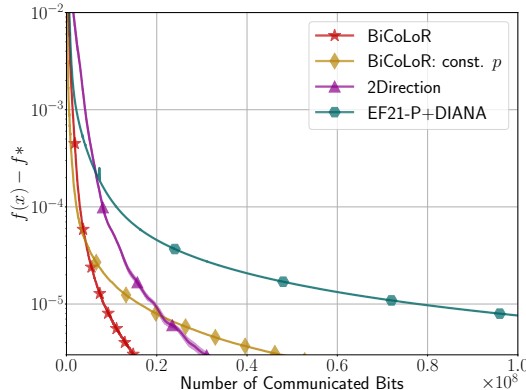

Figure 3: Logistic regression without regularization on the `w8a` dataset. The compression scheme combines sparsification with $K = 100$ and Natural Compression. BiCoLoR with decreasing $p_t$ (as defined in Theorem 4.4) outperforms the constant-$p$ variant in the long run.

The results are shown in Figure 3. We observe that BiCoLoR with decreasing $p_t$ outperforms the constant-$p$ version, as well as the other algorithms. Here too, our theoretical findings demonstrating acceleration are confirmed in practice, and BiCoLoR sets a new standard.

## C  PROOF OF THEOREM 4.1

We define the Euclidean space $\boldsymbol{\mathcal{X}} := \left(\mathbb{R}^d\right)^{n+2}$ endowed with the weighted inner product

$$\langle \mathbf{x}, \mathbf{x}' \rangle_{\boldsymbol{\mathcal{X}}} := \sum_{i=1}^n \langle x_i, x_i' \rangle + 2n \langle x_s, x_s' \rangle + n \langle y, y' \rangle,$$

for every $\mathbf{x} = (x_1, \ldots, x_n, x_s, y)$, $\mathbf{x}' = (x_1', \ldots, x_n', x_s', y')$. We also define the Euclidean spaces $\boldsymbol{\mathcal{U}} := \left(\mathbb{R}^d\right)^n$ endowed with the standard inner product $\langle \mathbf{u}, \mathbf{u}' \rangle_{\boldsymbol{\mathcal{U}}} := \sum_{i=1}^n \langle u_i, u_i' \rangle$, for every $\mathbf{u} = (u_1, \ldots, u_n)$, $\mathbf{u}' = (u_1', \ldots, u_n')$, and $\boldsymbol{\mathcal{U}}_y := \mathbb{R}^d$ endowed with the weighted inner product $\langle u, u' \rangle_{\boldsymbol{\mathcal{U}}_y} := n \langle u, u' \rangle$, for every $u, u' \in \mathbb{R}^d$.

We reformulate the problem (3) as

$$\min_{\mathbf{x} \in \boldsymbol{\mathcal{X}}} \mathbf{f}(\mathbf{x}) \quad \text{s.t.} \quad D\mathbf{x} = \mathbf{0} \text{ and } D_y \mathbf{x} = \mathbf{0}, \tag{15}$$

where

$$\mathbf{f} : \mathbf{x} = (x_1, \ldots, x_n, x_s, y) \in \boldsymbol{\mathcal{X}} \mapsto \sum_{i=1}^n f_i(x_i) + 2n f_s(x_s) + n g(y),$$

$$D : \boldsymbol{\mathcal{X}} \to \boldsymbol{\mathcal{U}} : (x_1, \ldots, x_n, x_s, y) \mapsto (x_1 - x_s, \ldots, x_n - x_s),$$

$$D_y : \boldsymbol{\mathcal{X}} \to \boldsymbol{\mathcal{U}}_y : (x_1, \ldots, x_n, x_s, y) \mapsto (y - x_s),$$

and $\mathbf{0}$ denotes the zero element of the Euclidean space.

We note that the function $\mathbf{f}$ in $\boldsymbol{\mathcal{X}}$ is $L$-smooth and $\mu$-strongly convex, and $\nabla \mathbf{f}(\mathbf{x}) = \left( \nabla f_1(x_1), \ldots \nabla f_n(x_n), \nabla f_s(x_s), \nabla g(y) \right)$. The adjoint operators are

$$D^* : \boldsymbol{\mathcal{U}} \to \boldsymbol{\mathcal{X}} : (u_1, \ldots, u_n) \mapsto \left( u_1, \ldots, u_n, -\frac{1}{2n} \sum_{i=1}^n u_i, 0 \right),$$

$$D_y^* : \boldsymbol{\mathcal{U}}_y \to \boldsymbol{\mathcal{X}} : u_y \mapsto (0, \ldots, 0, -\frac{1}{2} u_y, u_y).$$

We also introduce vector notations for the variables in BiCoLoR. We define $\mathbf{x}^\star := (x^\star, \ldots, x^\star, x^\star, x^\star) \in \boldsymbol{\mathcal{X}}$ as the unique solution to (15), $\mathbf{w}^\star := \mathbf{x}^\star - \gamma \nabla \mathbf{f}(\mathbf{x}^\star)$ and, for every $t \geq 0$,

$\mathbf{x}^t := (x_1^t, \ldots, x_n^t, x_s^t, y^t)$, $\hat{\mathbf{x}}^t := (\hat{x}_1^t, \ldots, \hat{x}_n^t, \hat{x}_s^t, \hat{y}^t)$, $\mathbf{u}^t := (u_1^t, \ldots, u_n^t)$, $\mathbf{u}^\star := (u_1^\star, \ldots, u_n^\star) = \nabla\mathbf{f}(\mathbf{x}^\star)$, $\mathbf{w}^t := \mathbf{x}^t - \gamma\nabla\mathbf{f}(\mathbf{x}^t)$, and the $\sigma$-algebra $\mathcal{F}^t$ generated by the collection of random variables $\mathbf{x}^0, \mathbf{u}^0, u_y^0 \ldots, \mathbf{x}^t, \mathbf{u}^t, u_y^t$.

We first consider the case $k = d$, so that $\Omega^t = [d]$ and full vectors are compressed.

Let $t \geq 0$. We can write the iteration of BiCoLoR as

$$
\begin{cases}
\hat{\mathbf{x}}^t := \mathbf{x}^t - \gamma\nabla\mathbf{f}(\mathbf{x}^t) + \gamma D^*\mathbf{u}^t + \gamma D_y^* u_y^t = \mathbf{w}^t + \gamma D^*\mathbf{u}^t + \gamma D_y^* u_y^t \\
\text{flip a coin } \theta^t \in \{0,1\}, \text{ with } \mathrm{Prob}(\theta^t = 1) = p \\
\textbf{if } \theta^t = 1 \\
\quad \mathbf{r}^t := \left(\hat{x}_1^t - c_s^t - \hat{y}^t, \ldots, \hat{x}_n^t - c_s^t - \hat{y}^t, \frac{1}{2}(\hat{x}_s^t - \bar{c}^t - \hat{y}^t), 0\right) \\
\quad \mathbf{r}_y^t := \left(0, \ldots, 0, \frac{1}{2}(\hat{x}_s^t - \hat{y}^t), -c_s^t\right) \\
\quad \mathbf{x}^{t+1} := \hat{\mathbf{x}}^t - \rho\mathbf{r} - \rho_y\mathbf{r}_y \\
\quad \mathbf{u}^{t+1} := \mathbf{u}^t - \frac{p\eta}{\gamma}(c_1^t - c_s^t, \ldots, c_n^t - c_s^t) \\
\quad u_y^{t+1} := u_y^t + \frac{p\eta_y}{\gamma}c_s^t \\
\textbf{else} \\
\quad \mathbf{x}^{t+1} := \hat{\mathbf{x}}^t \\
\quad \mathbf{u}^{t+1} := \mathbf{u}^t, u_y^{t+1} = u_y^t \\
\textbf{end if}
\end{cases}
\tag{16}
$$

We have

$$
\mathbb{E}\left[\mathbf{x}^{t+1} \mid \mathcal{F}^t, \theta = 1\right] = \hat{\mathbf{x}}^t - \rho D^* D\hat{\mathbf{x}}^t - \rho_y D_y^* D_y \hat{\mathbf{x}}^t.
\tag{17}
$$

That is, $\mathbf{x}^{t+1}$ is updated using a stochastic unbiased estimate of the right hand side of (17) made from the compressed vectors, because the server does not know the $\hat{x}_i^t$ and the clients do not know $\hat{x}_s^t$, the only available information consists of the compressed vectors. Similarly,

$$
\mathbb{E}\left[\mathbf{u}^{t+1} \mid \mathcal{F}^t, \theta = 1\right] = \mathbf{u}^t - \frac{p\eta}{\gamma}D\hat{\mathbf{x}}^t
$$

$$
\mathbb{E}\left[u_y^{t+1} \mid \mathcal{F}^t, \theta = 1\right] = u_y^t - \frac{p\eta_y}{\gamma}D_y\hat{\mathbf{x}}^t.
$$

Thus,

$$
\begin{aligned}
\mathbb{E}\left[\left\|\mathbf{x}^{t+1} - \mathbf{x}^\star\right\|_{\boldsymbol{\mathcal{X}}}^2 \mid \mathcal{F}^t\right] &= (1-p)\left\|\hat{\mathbf{x}}^t - \mathbf{x}^\star\right\|_{\boldsymbol{\mathcal{X}}}^2 + p\mathbb{E}\left[\left\|\mathbf{x}^{t+1} - \mathbf{x}^\star\right\|_{\boldsymbol{\mathcal{X}}}^2 \mid \mathcal{F}^t, \theta^t = 1\right] \\
&= (1-p)\left\|\hat{\mathbf{x}}^t - \mathbf{x}^\star\right\|_{\boldsymbol{\mathcal{X}}}^2 + p\left\|\hat{\mathbf{x}}^t - \hat{\mathbf{x}}^\star - \rho D^* D\hat{\mathbf{x}}^t - \rho_y D_y^* D_y \hat{\mathbf{x}}^t\right\|_{\boldsymbol{\mathcal{X}}}^2 \\
&\quad + pn(\rho^2 + \rho_y^2)\mathbb{E}\left[\left\|c_s^t - (\hat{x}_s^t - \hat{y}^t)\right\|^2 \mid \mathcal{F}^t, \theta^t = 1\right] \\
&\quad + p\frac{2n\rho^2}{4}\mathbb{E}\left[\left\|\bar{c}^t - \frac{1}{n}\sum_{i=1}^n(\hat{x}_i^t - \hat{y}^t)\right\|^2 \mid \mathcal{F}^t, \theta^t = 1\right] \\
&\leq (1-p)\left\|\hat{\mathbf{x}}^t - \mathbf{x}^\star\right\|_{\boldsymbol{\mathcal{X}}}^2 + p\left\|\hat{\mathbf{x}}^t - \hat{\mathbf{x}}^\star\right\|_{\boldsymbol{\mathcal{X}}}^2 \\
&\quad + p\left\|\rho D^* D\hat{\mathbf{x}}^t + \rho_y D_y^* D_y \hat{\mathbf{x}}^t\right\|_{\boldsymbol{\mathcal{X}}}^2 \\
&\quad - 2p\left\langle\hat{\mathbf{x}}^t - \hat{\mathbf{x}}^\star, \rho D^* D\hat{\mathbf{x}}^t + \rho_y D_y^* D_y \hat{\mathbf{x}}^t\right\rangle_{\boldsymbol{\mathcal{X}}} \\
&\quad + pn(\rho^2 + \rho_y^2)\omega_s\left\|\hat{x}_s^t - \hat{y}^t\right\|^2 + pn\rho^2\frac{\omega_{\mathrm{av}}}{2n}\sum_{i=1}^n\left\|\hat{x}_i^t - \hat{y}^t\right\|^2 \\
&= \left\|\hat{\mathbf{x}}^t - \mathbf{x}^\star\right\|_{\boldsymbol{\mathcal{X}}}^2 + p\left\|\rho D^* D\hat{\mathbf{x}}^t + \rho_y D_y^* D_y \hat{\mathbf{x}}^t\right\|_{\boldsymbol{\mathcal{X}}}^2 \\
&\quad - 2p\rho\left\|D\hat{\mathbf{x}}^t\right\|_{\boldsymbol{\mathcal{U}}}^2 - 2p\rho_y\left\|D_y\hat{\mathbf{x}}^t\right\|_{\boldsymbol{\mathcal{U}}_y}^2 \\
&\quad + pn(\rho^2 + \rho_y^2)\omega_s\left\|\hat{x}_s^t - \hat{y}^t\right\|^2 + \frac{p\rho^2\omega_{\mathrm{av}}}{2}\sum_{i=1}^n\left\|\hat{x}_i^t - \hat{y}^t\right\|^2.
\end{aligned}
$$

Moreover,

$$
\begin{aligned}
\left\|\hat{\mathbf{x}}^t - \mathbf{x}^\star\right\|_{\boldsymbol{\mathcal{X}}}^2 &= \left\|\mathbf{w}^t - \mathbf{w}^\star\right\|_{\boldsymbol{\mathcal{X}}}^2 + \gamma^2\left\|D^*(\mathbf{u}^t - \mathbf{u}^\star) + D_y^*(u_y^t - u_y^\star)\right\|_{\boldsymbol{\mathcal{X}}}^2 \\
&\quad + 2\gamma\left\langle\mathbf{w}^t - \mathbf{w}^\star, D^*(\mathbf{u}^t - \mathbf{u}^\star) + D_y^*(u_y^t - u_y^\star)\right\rangle_{\boldsymbol{\mathcal{X}}}.
\end{aligned}
$$

On the other hand,

$$\mathbb{E}\Big[\big\|\mathbf{u}^{t+1} - \mathbf{u}^\star\big\|_{\boldsymbol{\mathcal{U}}}^2 \mid \mathcal{F}^t\Big] = (1-p)\big\|\mathbf{u}^t - \mathbf{u}^\star\big\|_{\boldsymbol{\mathcal{U}}}^2 + p\mathbb{E}\Big[\big\|\mathbf{u}^{t+1} - \mathbf{u}^\star\big\|_{\boldsymbol{\mathcal{U}}}^2 \mid \mathcal{F}^t, \theta^t = 1\Big]$$

$$= (1-p)\big\|\mathbf{u}^t - \mathbf{u}^\star\big\|_{\boldsymbol{\mathcal{U}}}^2 + p\left\|\mathbf{u}^t - \mathbf{u}^\star - \frac{p\eta}{\gamma}D\hat{\mathbf{x}}^t\right\|_{\boldsymbol{\mathcal{U}}}^2$$

$$+ \frac{p^3\eta^2}{\gamma^2}\mathbb{E}\left[\sum_{i=1}^n \big\|c_i^t - c_s^t - (\hat{x}_i^t - \hat{x}_s^t)\big\|^2 \mid \mathcal{F}^t, \theta^t = 1\right].$$

Let $i \in [n]$. Since $\mathcal{C}_i$ and $\mathcal{C}_s$ are supposed independent, we have

$$\mathbb{E}\Big[\big\|c_i^t - c_s^t - (\hat{x}_i^t - \hat{x}_s^t)\big\|^2 \mid \mathcal{F}^t, \theta^t = 1\Big] = \mathbb{E}\Big[\big\|c_i^t - (\hat{x}_i^t - \hat{y}^t)\big\|^2 \mid \mathcal{F}^t, \theta^t = 1\Big]$$

$$+ \mathbb{E}\Big[\big\|c_s^t - (\hat{x}_s^t - \hat{y}^t)\big\|^2 \mid \mathcal{F}^t, \theta^t = 1\Big]$$

$$\leq \omega\big\|\hat{x}_i^t - \hat{y}^t\big\|^2 + \omega_s\big\|\hat{x}_s^t - \hat{y}^t\big\|^2.$$

Therefore,

$$\mathbb{E}\Big[\big\|\mathbf{u}^{t+1} - \mathbf{u}^\star\big\|_{\boldsymbol{\mathcal{U}}}^2 \mid \mathcal{F}^t\Big] \leq (1-p)\big\|\mathbf{u}^t - \mathbf{u}^\star\big\|_{\boldsymbol{\mathcal{U}}}^2 + p\big\|\mathbf{u}^t - \mathbf{u}^\star\big\|_{\boldsymbol{\mathcal{U}}}^2 + \frac{p^3\eta^2}{\gamma^2}\big\|D\hat{\mathbf{x}}^t\big\|_{\boldsymbol{\mathcal{U}}}^2$$

$$- \frac{2p^2\eta}{\gamma}\langle\mathbf{u}^t - \mathbf{u}^\star, D\hat{\mathbf{x}}^t\rangle_{\boldsymbol{\mathcal{U}}}$$

$$+ \frac{p^3\eta^2\omega}{\gamma^2}\sum_{i=1}^n\big\|\hat{x}_i^t - \hat{y}^t\big\|^2 + \frac{p^3\eta^2\omega_s n}{\gamma^2}\big\|\hat{x}_s^t - \hat{y}^t\big\|^2.$$

Moreover,

$$\mathbb{E}\Big[\big\|u_y^{t+1} - u_y^\star\big\|_{\boldsymbol{\mathcal{U}}_y}^2 \mid \mathcal{F}^t\Big] = (1-p)\big\|\mathbf{u}^t - \mathbf{u}^\star\big\|_{\boldsymbol{\mathcal{U}}_y}^2 + p\mathbb{E}\Big[\big\|\mathbf{u}^{t+1} - \mathbf{u}^\star\big\|_{\boldsymbol{\mathcal{U}}_y}^2 \mid \mathcal{F}^t, \theta^t = 1\Big]$$

$$= (1-p)\big\|u_y^t - u_y^\star\big\|_{\boldsymbol{\mathcal{U}}_y}^2 + p\left\|u_y^t - u_y^\star - \frac{p\eta_y}{\gamma}D_y\hat{\mathbf{x}}^t\right\|_{\boldsymbol{\mathcal{U}}_y}^2$$

$$+ \frac{p^3\eta_y^2 n}{\gamma^2}\mathbb{E}\Big[\big\|c_s^t - (\hat{x}_s^t - \hat{y}^t)\big\|^2 \mid \mathcal{F}^t, \theta^t = 1\Big]$$

$$\leq (1-p)\big\|u_y^t - u_y^\star\big\|_{\boldsymbol{\mathcal{U}}_y}^2 + p\big\|u_y^t - u_y^\star\big\|_{\boldsymbol{\mathcal{U}}_y}^2 + \frac{p^3\eta_y^2}{\gamma^2}\big\|D_y\hat{\mathbf{x}}^t\big\|_{\boldsymbol{\mathcal{U}}_y}^2$$

$$- \frac{2p^2\eta_y}{\gamma}\langle u_y^t - u_y^\star, D_y\hat{\mathbf{x}}^t\rangle_{\boldsymbol{\mathcal{U}}_y} + \frac{p^3\eta_y^2\omega_s n}{\gamma^2}\big\|\hat{x}_s^t - \hat{y}^t\big\|^2.$$

Thus, using the fact that $D\mathbf{x}^\star = D_y\mathbf{x}^\star = \mathbf{0}$, we have

$$\frac{\gamma}{p^2\eta}\mathbb{E}\Big[\big\|\mathbf{u}^{t+1} - \mathbf{u}^\star\big\|_{\boldsymbol{\mathcal{U}}}^2 \mid \mathcal{F}^t\Big] + \frac{\gamma}{p^2\eta_y}\mathbb{E}\Big[\big\|u_y^{t+1} - u_y^\star\big\|_{\boldsymbol{\mathcal{U}}_y}^2 \mid \mathcal{F}^t\Big]$$

$$\leq \frac{\gamma}{p^2\eta}\big\|\mathbf{u}^t - \mathbf{u}^\star\big\|_{\boldsymbol{\mathcal{U}}}^2 + \frac{\gamma}{p^2\eta_y}\big\|u_y^t - u_y^\star\big\|_{\boldsymbol{\mathcal{U}}_y}^2 - 2\big\langle D^*(\mathbf{u}^t - \mathbf{u}^\star) + D_y^*(u_y^t - u_y^\star), \hat{\mathbf{x}}^t - \mathbf{x}^\star\big\rangle_{\boldsymbol{\mathcal{X}}}$$

$$+ \frac{p\eta}{\gamma}\big\|D\hat{\mathbf{x}}^t\big\|_{\boldsymbol{\mathcal{U}}}^2 + \frac{p\eta_y}{\gamma}\big\|D_y\hat{\mathbf{x}}^t\big\|_{\boldsymbol{\mathcal{U}}_y}^2 + \frac{p\eta\omega}{\gamma}\sum_{i=1}^n\big\|\hat{x}_i^t - \hat{y}^t\big\|^2 + \frac{p(\eta + \eta_y)\omega_s}{\gamma}\big\|D_y\hat{\mathbf{x}}^t\big\|_{\boldsymbol{\mathcal{U}}_y}^2$$

$$= \frac{\gamma}{p^2\eta}\big\|\mathbf{u}^t - \mathbf{u}^\star\big\|_{\boldsymbol{\mathcal{U}}}^2 + \frac{\gamma}{p^2\eta_y}\big\|u_y^t - u_y^\star\big\|_{\boldsymbol{\mathcal{U}}_y}^2 - 2\gamma\big\|D^*(\mathbf{u}^t - \mathbf{u}^\star) + D_y^*(u_y^t - u_y^\star)\big\|_{\boldsymbol{\mathcal{X}}}^2$$

$$- 2\big\langle D^*(\mathbf{u}^t - \mathbf{u}^\star) + D_y^*(u_y^t - u_y^\star), \mathbf{w}^t - \mathbf{w}^\star\big\rangle_{\boldsymbol{\mathcal{X}}}$$

$$+ \frac{p\eta}{\gamma}\big\|D\hat{\mathbf{x}}^t\big\|_{\boldsymbol{\mathcal{U}}}^2 + \frac{p\eta_y}{\gamma}\big\|D_y\hat{\mathbf{x}}^t\big\|_{\boldsymbol{\mathcal{U}}_y}^2 + \frac{p\eta\omega}{\gamma}\sum_{i=1}^n\big\|\hat{x}_i^t - \hat{y}^t\big\|^2 + \frac{p(\eta + \eta_y)\omega_s}{\gamma}\big\|D_y\hat{\mathbf{x}}^t\big\|_{\boldsymbol{\mathcal{U}}_y}^2.$$

Hence,

$$\frac{1}{\gamma}\mathbb{E}\Big[\left\|\mathbf{x}^{t+1}-\mathbf{x}^{\star}\right\|_{\boldsymbol{\mathcal{X}}}^{2}\mid\mathcal{F}^{t}\Big]+\frac{\gamma}{p^{2}\eta}\mathbb{E}\Big[\left\|\mathbf{u}^{t+1}-\mathbf{u}^{\star}\right\|_{\boldsymbol{\mathcal{U}}}^{2}\mid\mathcal{F}^{t}\Big]+\frac{\gamma}{p^{2}\eta_{y}}\mathbb{E}\Big[\left\|u_{y}^{t+1}-u_{y}^{\star}\right\|_{\boldsymbol{\mathcal{U}}_{y}}^{2}\mid\mathcal{F}^{t}\Big]$$

$$\leq\frac{1}{\gamma}\left\|\mathbf{w}^{t}-\mathbf{w}^{\star}\right\|_{\boldsymbol{\mathcal{X}}}^{2}+\gamma\left\|D^{*}(\mathbf{u}^{t}-\mathbf{u}^{\star})+D_{y}^{*}(u_{y}^{t}-u_{y}^{\star})\right\|_{\boldsymbol{\mathcal{X}}}^{2}$$

$$+2\big\langle\mathbf{w}^{t}-\mathbf{w}^{\star},D^{*}(\mathbf{u}^{t}-\mathbf{u}^{\star})+D_{y}^{*}(u_{y}^{t}-u_{y}^{\star})\big\rangle_{\boldsymbol{\mathcal{X}}}$$

$$+\frac{\gamma}{p^{2}\eta}\left\|\mathbf{u}^{t}-\mathbf{u}^{\star}\right\|_{\boldsymbol{\mathcal{U}}}^{2}+\frac{\gamma}{p^{2}\eta_{y}}\left\|u_{y}^{t}-u_{y}^{\star}\right\|_{\boldsymbol{\mathcal{U}}_{y}}^{2}-2\gamma\left\|D^{*}(\mathbf{u}^{t}-\mathbf{u}^{\star})+D_{y}^{*}(u_{y}^{t}-u_{y}^{\star})\right\|_{\boldsymbol{\mathcal{X}}}^{2}$$

$$-2\big\langle D^{*}(\mathbf{u}^{t}-\mathbf{u}^{\star})+D_{y}^{*}(u_{y}^{t}-u_{y}^{\star}),\mathbf{w}^{t}-\mathbf{w}^{\star}\big\rangle_{\boldsymbol{\mathcal{X}}}$$

$$+\frac{p}{\gamma}\left\|\rho D^{*}D\hat{\mathbf{x}}^{t}+\rho_{y}D_{y}^{*}D_{y}\hat{\mathbf{x}}^{t}\right\|_{\boldsymbol{\mathcal{X}}}^{2}-\frac{2p\rho}{\gamma}\left\|D\hat{\mathbf{x}}^{t}\right\|_{\boldsymbol{\mathcal{U}}}^{2}-\frac{2p\rho_{y}}{\gamma}\left\|D_{y}\hat{\mathbf{x}}^{t}\right\|_{\boldsymbol{\mathcal{U}}_{y}}^{2}$$

$$+\frac{pn(\rho^{2}+\rho_{y}^{2})\omega_{s}}{\gamma}\left\|\hat{x}_{s}^{t}-\hat{y}^{t}\right\|^{2}+\frac{p\rho^{2}\omega_{\mathrm{av}}}{2\gamma}\sum_{i=1}^{n}\left\|\hat{x}_{i}^{t}-\hat{y}^{t}\right\|^{2}$$

$$+\frac{p\eta}{\gamma}\left\|D\hat{\mathbf{x}}^{t}\right\|_{\boldsymbol{\mathcal{U}}}^{2}+\frac{p\eta_{y}}{\gamma}\left\|D_{y}\hat{\mathbf{x}}^{t}\right\|_{\boldsymbol{\mathcal{U}}_{y}}^{2}+\frac{p\eta\omega}{\gamma}\sum_{i=1}^{n}\left\|\hat{x}_{i}^{t}-\hat{y}^{t}\right\|^{2}+\frac{p(\eta+\eta_{y})\omega_{s}}{\gamma}\left\|D_{y}\hat{\mathbf{x}}^{t}\right\|_{\boldsymbol{\mathcal{U}}_{y}}^{2}$$

$$=\frac{1}{\gamma}\left\|\mathbf{w}^{t}-\mathbf{w}^{\star}\right\|_{\boldsymbol{\mathcal{X}}}^{2}+\frac{\gamma}{p^{2}\eta}\left\|\mathbf{u}^{t}-\mathbf{u}^{\star}\right\|_{\boldsymbol{\mathcal{U}}}^{2}+\frac{\gamma}{p^{2}\eta_{y}}\left\|u_{y}^{t}-u_{y}^{\star}\right\|_{\boldsymbol{\mathcal{U}}_{y}}^{2}$$

$$-\gamma\left\|D^{*}(\mathbf{u}^{t}-\mathbf{u}^{\star})+D_{y}^{*}(u_{y}^{t}-u_{y}^{\star})\right\|_{\boldsymbol{\mathcal{X}}}^{2}+\frac{p}{\gamma}\left\|\rho D^{*}D\hat{\mathbf{x}}^{t}+\rho_{y}D_{y}^{*}D_{y}\hat{\mathbf{x}}^{t}\right\|_{\boldsymbol{\mathcal{X}}}^{2}$$

$$+\frac{p\eta-2p\rho}{\gamma}\left\|D\hat{\mathbf{x}}^{t}\right\|_{\boldsymbol{\mathcal{U}}}^{2}+\frac{p\rho^{2}\omega_{\mathrm{av}}+2p\eta\omega}{2\gamma}\sum_{i=1}^{n}\left\|\hat{x}_{i}^{t}-\hat{y}^{t}\right\|^{2}$$

$$+\frac{p\eta_{y}-2p\rho_{y}+p(\eta+\eta_{y})\omega_{s}+p(\rho^{2}+\rho_{y}^{2})\omega_{s}}{\gamma}\left\|D_{y}\hat{\mathbf{x}}^{t}\right\|_{\boldsymbol{\mathcal{U}}_{y}}^{2}.$$

Using Young's inequality, we have

$$\sum_{i=1}^{n}\left\|\hat{x}_{i}^{t}-\hat{y}^{t}\right\|^{2}\leq\sum_{i=1}^{n}\Big(2\left\|\hat{x}_{i}^{t}-\hat{x}_{s}^{t}\right\|^{2}+2\left\|\hat{x}_{s}^{t}-\hat{y}^{t}\right\|^{2}\Big)$$

$$=2\left\|D\hat{\mathbf{x}}^{t}\right\|_{\boldsymbol{\mathcal{U}}}^{2}+2\left\|D_{y}\hat{\mathbf{x}}^{t}\right\|_{\boldsymbol{\mathcal{U}}_{y}}^{2}.$$

The leading eigenvalue of $D^{*}D+D_{y}^{*}D_{y}$ is 2, with corresponding eigenvector $(x,\ldots,x,-x,x)$ for any $x\in\mathbb{R}^{d}$, so that

$$\left\|\rho D^{*}D\hat{\mathbf{x}}^{t}+\rho_{y}D_{y}^{*}D_{y}\hat{\mathbf{x}}^{t}\right\|_{\boldsymbol{\mathcal{X}}}^{2}\leq\left\|D^{*}D+D_{y}^{*}D_{y}\right\|\Big(\left\|\rho D\hat{\mathbf{x}}^{t}\right\|_{\boldsymbol{\mathcal{U}}}^{2}+\left\|\rho_{y}D_{y}\hat{\mathbf{x}}^{t}\right\|_{\boldsymbol{\mathcal{U}}_{y}}^{2}\Big)$$

$$=2\rho^{2}\left\|D\hat{\mathbf{x}}^{t}\right\|_{\boldsymbol{\mathcal{U}}}^{2}+2\rho_{y}^{2}\left\|D_{y}\hat{\mathbf{x}}^{t}\right\|_{\boldsymbol{\mathcal{U}}_{y}}^{2}.$$

Therefore,

$$\frac{1}{\gamma}\mathbb{E}\Big[\left\|\mathbf{x}^{t+1}-\mathbf{x}^{\star}\right\|_{\boldsymbol{\mathcal{X}}}^{2}\mid\mathcal{F}^{t}\Big]+\frac{\gamma}{p^{2}\eta}\mathbb{E}\Big[\left\|\mathbf{u}^{t+1}-\mathbf{u}^{\star}\right\|_{\boldsymbol{\mathcal{U}}}^{2}\mid\mathcal{F}^{t}\Big]+\frac{\gamma}{p^{2}\eta_{y}}\mathbb{E}\Big[\left\|u_{y}^{t+1}-u_{y}^{\star}\right\|_{\boldsymbol{\mathcal{U}}_{y}}^{2}\mid\mathcal{F}^{t}\Big]$$

$$\leq\frac{1}{\gamma}\left\|\mathbf{w}^{t}-\mathbf{w}^{\star}\right\|_{\boldsymbol{\mathcal{X}}}^{2}+\frac{\gamma}{p^{2}\eta}\left\|\mathbf{u}^{t}-\mathbf{u}^{\star}\right\|_{\boldsymbol{\mathcal{U}}}^{2}+\frac{\gamma}{p^{2}\eta_{y}}\left\|u_{y}^{t}-u_{y}^{\star}\right\|_{\boldsymbol{\mathcal{U}}_{y}}^{2}$$

$$-\gamma\left\|D^{*}(\mathbf{u}^{t}-\mathbf{u}^{\star})+D_{y}^{*}(u_{y}^{t}-u_{y}^{\star})\right\|_{\boldsymbol{\mathcal{X}}}^{2}$$

$$+\frac{-2p\rho+p\rho^{2}(2+\omega_{\mathrm{av}})+p\eta(1+2\omega)}{\gamma}\left\|D\hat{\mathbf{x}}^{t}\right\|_{\boldsymbol{\mathcal{U}}}^{2}$$

$$+\frac{-2p\rho_{y}+p\rho_{y}^{2}(2+\omega_{s})+p\rho^{2}(\omega_{s}+\omega_{\mathrm{av}})+p\eta(\omega_{s}+2\omega)+p\eta_{y}(1+\omega_{s})}{\gamma}\left\|D_{y}\hat{\mathbf{x}}^{t}\right\|_{\boldsymbol{\mathcal{U}}_{y}}^{2}.$$

We need to choose $\rho, \rho_y, \eta, \eta_y$ small enough to remove the last two squared norm terms. We choose

$$\rho = \rho_y = \frac{1}{2 + \omega_{\text{av}} + 2\omega_s},$$

$$\eta = \eta_y = \frac{1}{(1 + 2\omega + 2\omega_s)(2 + \omega_{\text{av}} + 2\omega_s)}.$$

This way,

$$-2p\rho_y + p\rho_y^2(2 + \omega_s) + p\rho^2(\omega_s + \omega_{\text{av}}) + p\eta(\omega_s + 2\omega) + p\eta_y(1 + \omega_s) = 0$$

and

$$-2p\rho + p\rho^2(2 + \omega_{\text{av}}) + p\eta(1 + 2\omega) \le -2p\rho + p\rho^2(2 + \omega_{\text{av}} + 2\omega_s) + p\eta(1 + 2\omega + 2\omega_s) = 0.$$

Then, according to Condat & Richtárik (2023, Lemma 1),

$$\left\| \mathbf{w}^t - \mathbf{w}^\star \right\|_{\boldsymbol{\mathcal{X}}}^2 = \left\| (\text{Id} - \gamma\nabla\mathbf{f})\mathbf{x}^t - (\text{Id} - \gamma\nabla\mathbf{f})\mathbf{x}^\star \right\|_{\boldsymbol{\mathcal{X}}}^2$$

$$\le \max(1 - \gamma\mu, \gamma L - 1)^2 \left\| \mathbf{x}^t - \mathbf{x}^\star \right\|_{\boldsymbol{\mathcal{X}}}^2. \tag{18}$$

Moreover, we define the concatenated operator $D_c : \mathbf{x} \in \boldsymbol{\mathcal{X}} \mapsto (D\mathbf{x}, D_y\mathbf{x}) \in \boldsymbol{\mathcal{U}} \times \boldsymbol{\mathcal{U}}_y$. $D_c$ has full range, since for every $(x_1, \ldots, x_n, y) \in \boldsymbol{\mathcal{U}} \times \boldsymbol{\mathcal{U}}_y$, $D_c(x_1, \ldots, x_n, 0, y) = (x_1, \ldots, x_n, y)$. Equivalently, $D_c^*$ is injective. For every $(\mathbf{u}, u_y) \in \boldsymbol{\mathcal{U}} \times \boldsymbol{\mathcal{U}}_y$, we have $\left\| D^*\mathbf{u} + D_y^* u_y \right\|_{\boldsymbol{\mathcal{X}}}^2 = \left\| D_c^*(\mathbf{u}, u_y) \right\|_{\boldsymbol{\mathcal{X}}}^2 \ge \lambda_{\min}(D_c D_c^*)\left( \|\mathbf{u}\|_{\boldsymbol{\mathcal{U}}}^2 + \|u_y\|_{\boldsymbol{\mathcal{U}}_y}^2 \right)$, where $\lambda_{\min}(D_c D_c^*)$ is the smallest eigenvalue of $D_c D_c^*$, which is positive because $D_c^*$ is injective. This eigenvalue is 1, with corresponding eigenvectors of the form $(u_1, \ldots, u_n, -\frac{1}{n}\sum_{i=1}^n u_i)$, that form a space of dimension $n$ (the $(n+1)$-th eigenvalue is 2, as mentioned above). Therefore,

$$\left\| D^*(\mathbf{u}^t - \mathbf{u}^\star) + D_y^*(u_y^t - u_y^\star) \right\|_{\boldsymbol{\mathcal{X}}}^2 \ge \left\| \mathbf{u}^t - \mathbf{u}^\star \right\|_{\boldsymbol{\mathcal{U}}}^2 + \left\| u_y^t - u_y^\star \right\|_{\boldsymbol{\mathcal{U}}_y}^2.$$

Hence,

$$\frac{1}{\gamma}\mathbb{E}\Big[ \left\| \mathbf{x}^{t+1} - \mathbf{x}^\star \right\|_{\boldsymbol{\mathcal{X}}}^2 \mid \mathcal{F}^t \Big] + \frac{\gamma}{p^2\eta}\mathbb{E}\Big[ \left\| \mathbf{u}^{t+1} - \mathbf{u}^\star \right\|_{\boldsymbol{\mathcal{U}}}^2 \mid \mathcal{F}^t \Big] + \frac{\gamma}{p^2\eta_y}\mathbb{E}\Big[ \left\| u_y^{t+1} - u_y^\star \right\|_{\boldsymbol{\mathcal{U}}_y}^2 \mid \mathcal{F}^t \Big]$$

$$\le \frac{1}{\gamma}\max(1 - \gamma\mu, \gamma L - 1)^2 \left\| \mathbf{x}^t - \mathbf{x}^\star \right\|_{\boldsymbol{\mathcal{X}}}^2 + \frac{\gamma}{p^2\eta}\left\| \mathbf{u}^t - \mathbf{u}^\star \right\|_{\boldsymbol{\mathcal{U}}}^2 + \frac{\gamma}{p^2\eta_y}\left\| u_y^t - u_y^\star \right\|_{\boldsymbol{\mathcal{U}}_y}^2$$

$$- \gamma\left\| \mathbf{u}^t - \mathbf{u}^\star \right\|_{\boldsymbol{\mathcal{U}}}^2 - \gamma\left\| u_y^t - u_y^\star \right\|_{\boldsymbol{\mathcal{U}}_y}^2$$

$$= \frac{1}{\gamma}\max(1 - \gamma\mu, \gamma L - 1)^2 \left\| \mathbf{x}^t - \mathbf{x}^\star \right\|_{\boldsymbol{\mathcal{X}}}^2 + \frac{\gamma}{p^2\eta}\big(1 - p^2\eta\big)\left\| \mathbf{u}^t - \mathbf{u}^\star \right\|_{\boldsymbol{\mathcal{U}}}^2$$

$$+ \frac{\gamma}{p^2\eta_y}\big(1 - p^2\eta_y\big)\left\| u_y^t - u_y^\star \right\|_{\boldsymbol{\mathcal{U}}_y}^2,$$

so that

$$\mathbb{E}\big[ \Psi^{t+1} \mid \mathcal{F}^t \big] \le \max\big( (1 - \gamma\mu)^2, (1 - \gamma L)^2, 1 - p^2\eta \big)\Psi^t. \tag{19}$$

Using the tower rule, we can unroll the recursion in (19) to obtain the unconditional expectation of $\Psi^{t+1}$.

Using classical results on supermartingale convergence (Bertsekas, 2015, Proposition A.4.5), it follows from (19) that $\Psi^t \to 0$ almost surely. Almost sure convergence of $\mathbf{x}^t, \mathbf{u}^t, u_y^t$ follows.

Finally, let us consider the general case $k \in [d]$. We observe that the analysis above is separable with respect to the $d$ coordinates of the vectors. From the perspective of a given coordinate $j \in [d]$, either the vector values at this coordinate are updated using communicated information, which happens if $\theta^t = 1$ and $j \in \Omega^t$, or they are updated using local information only, which happens if $\theta^t = 0$ or $j \notin \Omega^t$. So, since $j \in \Omega^t$ happens with probability $k/d$, the whole analysis above applies with $p$ replaced by $\frac{pk}{d}$.

# D  PROOF OF THEOREM 4.4

For every $t \geq 0$, we define $\mathcal{D}_{f_i}^t := \mathcal{D}_{f_i}(x_i^t, x^\star)$, for every $i \in [n]$, $\mathcal{D}_{f_s}^t := \mathcal{D}_{f_s}(x_s^t, x^\star)$, $\mathcal{D}_g^t := \mathcal{D}_g(y^t, x^\star)$.

We follow the same derivations as in Appendix C. However, instead of (18), we use

$$\left\|\mathbf{w}^t - \mathbf{w}^\star\right\|_{\boldsymbol{\mathcal{X}}}^2 = \left\|\mathbf{x}^t - \mathbf{x}^\star\right\|_{\boldsymbol{\mathcal{X}}}^2 - 2\gamma \left\langle \nabla\mathbf{f}(\mathbf{x}^t) - \nabla\mathbf{f}(\mathbf{x}^\star), \mathbf{x}^t - \mathbf{x}^\star\right\rangle_{\boldsymbol{\mathcal{X}}} + \gamma^2 \left\|\nabla\mathbf{f}(\mathbf{x}^t) - \nabla\mathbf{f}(\mathbf{x}^\star)\right\|_{\boldsymbol{\mathcal{X}}}^2. \tag{20}$$

Also, we choose

$$\rho = \rho_y = \frac{1}{2 + \omega_{\mathrm{av}} + 2\omega_s},$$

$$\eta = \eta_y < \frac{1}{(1 + 2\omega + 2\omega_s)(2 + \omega_{\mathrm{av}} + 2\omega_s)}$$

(for instance $\eta = \eta_y = \frac{0.99}{(1+2\omega+2\omega_s)(2+\omega_{\mathrm{av}}+2\omega_s)}$). This way,

$$\chi_y := 2\rho_y - \rho_y^2(2 + \omega_s) - \rho^2(\omega_s + \omega_{\mathrm{av}}) - \eta(\omega_s + 2\omega) - \eta_y(1 + \omega_s) > 0$$

and

$$\chi := 2\rho - \rho^2(2 + \omega_{\mathrm{av}}) - \eta(1 + 2\omega) \geq 2\rho - \rho^2(2 + \omega_{\mathrm{av}} + 2\omega_s) - \eta(1 + 2\omega + 2\omega_s) > 0.$$

Hence,

$$\frac{1}{\gamma}\mathbb{E}\left[\left\|\mathbf{x}^{t+1} - \mathbf{x}^\star\right\|_{\boldsymbol{\mathcal{X}}}^2 \mid \mathcal{F}^t\right] + \frac{\gamma}{p_{t+1}^2\eta}\mathbb{E}\left[\left\|\mathbf{u}^{t+1} - \mathbf{u}^\star\right\|_{\boldsymbol{\mathcal{U}}}^2 \mid \mathcal{F}^t\right] + \frac{\gamma}{p_{t+1}^2\eta}\mathbb{E}\left[\left\|u_y^{t+1} - u_y^\star\right\|_{\boldsymbol{\mathcal{U}}_y}^2 \mid \mathcal{F}^t\right]$$

$$\leq \frac{1}{\gamma}\left\|\mathbf{x}^t - \mathbf{x}^\star\right\|_{\boldsymbol{\mathcal{X}}}^2 - 2\left\langle \nabla\mathbf{f}(\mathbf{x}^t) - \nabla\mathbf{f}(\mathbf{x}^\star), \mathbf{x}^t - \mathbf{x}^\star\right\rangle_{\boldsymbol{\mathcal{X}}} + \gamma\left\|\nabla\mathbf{f}(\mathbf{x}^t) - \nabla\mathbf{f}(\mathbf{x}^\star)\right\|_{\boldsymbol{\mathcal{X}}}^2$$

$$+ \frac{\gamma}{p_{t+1}^2\eta}\left(1 - p_{t+1}^2\eta\right)\left\|\mathbf{u}^t - \mathbf{u}^\star\right\|_{\boldsymbol{\mathcal{U}}}^2 + \frac{\gamma}{p_{t+1}^2\eta}\left(1 - p_{t+1}^2\eta\right)\left\|u_y^t - u_y^\star\right\|_{\boldsymbol{\mathcal{U}}_y}^2$$

$$- \frac{p_{t+1}\chi}{\gamma}\left\|D\hat{\mathbf{x}}^t\right\|_{\boldsymbol{\mathcal{U}}}^2 - \frac{p_{t+1}\chi_y}{\gamma}\left\|D_y\hat{\mathbf{x}}^t\right\|_{\boldsymbol{\mathcal{U}}_y}^2.$$

So, assuming for simplicity that $\gamma \leq \frac{1}{L}$, we have

$$\frac{1}{\gamma}\mathbb{E}\left[\left\|\mathbf{x}^{t+1} - \mathbf{x}^\star\right\|_{\boldsymbol{\mathcal{X}}}^2 \mid \mathcal{F}^t\right] + \frac{\gamma}{p_{t+1}^2\eta}\mathbb{E}\left[\left\|\mathbf{u}^{t+1} - \mathbf{u}^\star\right\|_{\boldsymbol{\mathcal{U}}}^2 \mid \mathcal{F}^t\right] + \frac{\gamma}{p_{t+1}^2\eta}\mathbb{E}\left[\left\|u_y^{t+1} - u_y^\star\right\|_{\boldsymbol{\mathcal{U}}_y}^2 \mid \mathcal{F}^t\right]$$

$$\leq \frac{1}{\gamma}\left\|\mathbf{x}^t - \mathbf{x}^\star\right\|_{\boldsymbol{\mathcal{X}}}^2 - 2\sum_{i=1}^n \mathcal{D}_{f_i}^t - 4n\mathcal{D}_{f_s}^t - 2n\mathcal{D}_g^t + \left(\gamma - \frac{1}{L}\right)\left\|\nabla\mathbf{f}(\mathbf{x}^t) - \nabla\mathbf{f}(\mathbf{x}^\star)\right\|_{\boldsymbol{\mathcal{X}}}^2$$

$$- \frac{p_{t+1}\chi}{\gamma}\left\|D\hat{\mathbf{x}}^t\right\|_{\boldsymbol{\mathcal{U}}}^2 - \frac{p_{t+1}\chi_y}{\gamma}\left\|D_y\hat{\mathbf{x}}^t\right\|_{\boldsymbol{\mathcal{U}}_y}^2$$

$$+ \frac{\gamma}{p_{t+1}^2\eta}\left(1 - p_{t+1}^2\eta\right)\left\|\mathbf{u}^t - \mathbf{u}^\star\right\|_{\boldsymbol{\mathcal{U}}}^2 + \frac{\gamma}{p_{t+1}^2\eta}\left(1 - p_{t+1}^2\eta\right)\left\|u_y^t - u_y^\star\right\|_{\boldsymbol{\mathcal{U}}_y}^2$$

$$\leq \frac{1}{\gamma}\left\|\mathbf{x}^t - \mathbf{x}^\star\right\|_{\boldsymbol{\mathcal{X}}}^2 - 2\sum_{i=1}^n \mathcal{D}_{f_i}^t - 4n\mathcal{D}_{f_s}^t - 2n\mathcal{D}_g^t - \frac{p_{t+1}\chi}{\gamma}\left\|D\hat{\mathbf{x}}^t\right\|_{\boldsymbol{\mathcal{U}}}^2 - \frac{p_{t+1}\chi_y}{\gamma}\left\|D_y\hat{\mathbf{x}}^t\right\|_{\boldsymbol{\mathcal{U}}_y}^2$$

$$+ \gamma\left(\frac{1}{p_{t+1}^2\eta} - 1\right)\left\|\mathbf{u}^t - \mathbf{u}^\star\right\|_{\boldsymbol{\mathcal{U}}}^2 + \gamma\left(\frac{1}{p_{t+1}^2\eta} - 1\right)\left\|u_y^t - u_y^\star\right\|_{\boldsymbol{\mathcal{U}}_y}^2.$$

We choose

$$p_t = \sqrt{\frac{b}{a + t}}$$

for some $b \geq \frac{1}{\eta}$ and $a \geq b - 1$ (so that $p_t \in (0, 1]$ for every $t \geq 1$). Then we have

$$\frac{1}{p_{t+1}^2\eta} - 1 = \frac{a - b\eta + t + 1}{b\eta} \leq \frac{a + t}{b\eta} = \frac{1}{p_t^2\eta}.$$

Hence,

$$
\frac{1}{\gamma}\mathbb{E}\Big[\big\|\mathbf{x}^{t+1}-\mathbf{x}^\star\big\|_{\boldsymbol{\mathcal{X}}}^2 \mid \mathcal{F}^t\Big] + \frac{\gamma}{p_{t+1}^2\eta}\mathbb{E}\Big[\big\|\mathbf{u}^{t+1}-\mathbf{u}^\star\big\|_{\boldsymbol{\mathcal{U}}}^2 \mid \mathcal{F}^t\Big] + \frac{\gamma}{p_{t+1}^2\eta}\mathbb{E}\Big[\big\|u_y^{t+1}-u_y^\star\big\|_{\boldsymbol{\mathcal{U}}_y}^2 \mid \mathcal{F}^t\Big]
$$

$$
\le \frac{1}{\gamma}\big\|\mathbf{x}^t-\mathbf{x}^\star\big\|_{\boldsymbol{\mathcal{X}}}^2 - 2\sum_{i=1}^n \mathcal{D}_{f_i}^t - 4n\mathcal{D}_{f_s}^t - 2n\mathcal{D}_g^t - \frac{p_{t+1}\chi}{\gamma}\big\|D\hat{\mathbf{x}}^t\big\|_{\boldsymbol{\mathcal{U}}}^2 - \frac{p_{t+1}\chi_y}{\gamma}\big\|D_y\hat{\mathbf{x}}^t\big\|_{\boldsymbol{\mathcal{U}}_y}^2
$$

$$
+ \frac{\gamma}{p_t^2\eta}\big\|\mathbf{u}^t-\mathbf{u}^\star\big\|_{\boldsymbol{\mathcal{U}}}^2 + \frac{\gamma}{p_t^2\eta}\big\|u_y^t-u_y^\star\big\|_{\boldsymbol{\mathcal{U}}_y}^2 .
$$

By unrolling the recursion, we have, for every $t \ge 1$,

$$
\frac{1}{\gamma}\mathbb{E}\Big[\big\|\mathbf{x}^t-\mathbf{x}^\star\big\|_{\boldsymbol{\mathcal{X}}}^2 \mid \mathcal{F}^t\Big] + \frac{\gamma}{p_t^2\eta}\mathbb{E}\Big[\big\|\mathbf{u}^t-\mathbf{u}^\star\big\|_{\boldsymbol{\mathcal{U}}}^2 \mid \mathcal{F}^t\Big] + \frac{\gamma}{p_t^2\eta}\mathbb{E}\Big[\big\|u_y^t-u_y^\star\big\|_{\boldsymbol{\mathcal{U}}_y}^2 \mid \mathcal{F}^t\Big]
$$

$$
\le \frac{1}{\gamma}\big\|\mathbf{x}^0-\mathbf{x}^\star\big\|_{\boldsymbol{\mathcal{X}}}^2 + \frac{\gamma a}{\eta b}\big\|\mathbf{u}^0-\mathbf{u}^\star\big\|_{\boldsymbol{\mathcal{U}}}^2 + \frac{\gamma a}{\eta b}\big\|u_y^0-u_y^\star\big\|_{\boldsymbol{\mathcal{U}}_y}^2
$$

$$
- \sum_{t'=0}^{t-1}\mathbb{E}\left[ 2\sum_{i=1}^n \mathcal{D}_{f_i}^{t'} + 4n\mathcal{D}_{f_s}^{t'} + 2n\mathcal{D}_g^{t'} + \frac{p_{t'+1}\chi}{\gamma}\big\|D\hat{\mathbf{x}}^{t'}\big\|_{\boldsymbol{\mathcal{U}}}^2 + \frac{p_{t'+1}\chi_y}{\gamma}\big\|D_y\hat{\mathbf{x}}^{t'}\big\|_{\boldsymbol{\mathcal{U}}_y}^2 \right].
$$

This implies that

$$
\sum_{t=0}^\infty \mathbb{E}\left[ 2\sum_{i=1}^n \mathcal{D}_{f_i}^t + 4n\mathcal{D}_{f_s}^t + 2n\mathcal{D}_g^t + \frac{p_{t+1}\chi}{\gamma}\big\|D\hat{\mathbf{x}}^t\big\|_{\boldsymbol{\mathcal{U}}}^2 + \frac{p_{t+1}\chi_y}{\gamma}\big\|D_y\hat{\mathbf{x}}^t\big\|_{\boldsymbol{\mathcal{U}}_y}^2 \right]
$$

$$
\le \frac{1}{\gamma}\big\|\mathbf{x}^0-\mathbf{x}^\star\big\|_{\boldsymbol{\mathcal{X}}}^2 + \frac{\gamma a}{\eta b}\big\|\mathbf{u}^0-\mathbf{u}^\star\big\|_{\boldsymbol{\mathcal{U}}}^2 + \frac{\gamma a}{\eta b}\big\|u_y^0-u_y^\star\big\|_{\boldsymbol{\mathcal{U}}_y}^2 ,
$$

so that $\mathbb{E}\Big[2\sum_{i=1}^n \mathcal{D}_{f_i}^t + 4n\mathcal{D}_{f_s}^t + 2n\mathcal{D}_g^t + \frac{p_{t+1}\chi}{\gamma}\big\|D\hat{\mathbf{x}}^t\big\|_{\boldsymbol{\mathcal{U}}}^2 + \frac{p_{t+1}\chi_y}{\gamma}\big\|D_y\hat{\mathbf{x}}^t\big\|_{\boldsymbol{\mathcal{U}}_y}^2\Big] \to 0$ as $t \to +\infty$.
Moreover, for every $t \ge 0$,

$$
\mathbb{E}\Big[\big\|\mathbf{u}^t-\mathbf{u}^\star\big\|_{\boldsymbol{\mathcal{U}}}^2 \mid \mathcal{F}^t\Big] + \mathbb{E}\Big[\big\|u_y^t-u_y^\star\big\|_{\boldsymbol{\mathcal{U}}_y}^2 \mid \mathcal{F}^t\Big]
$$

$$
\le p_t^2\left( \frac{\eta}{\gamma^2}\big\|\mathbf{x}^0-\mathbf{x}^\star\big\|_{\boldsymbol{\mathcal{X}}}^2 + \frac{a}{b}\big\|\mathbf{u}^0-\mathbf{u}^\star\big\|_{\boldsymbol{\mathcal{U}}}^2 + \frac{a}{b}\big\|u_y^0-u_y^\star\big\|_{\boldsymbol{\mathcal{U}}_y}^2 \right)
$$

$$
\frac{b}{a+t}\left( \frac{\eta}{\gamma^2}\big\|\mathbf{x}^0-\mathbf{x}^\star\big\|_{\boldsymbol{\mathcal{X}}}^2 + \frac{a}{b}\big\|\mathbf{u}^0-\mathbf{u}^\star\big\|_{\boldsymbol{\mathcal{U}}}^2 + \frac{a}{b}\big\|u_y^0-u_y^\star\big\|_{\boldsymbol{\mathcal{U}}_y}^2 \right)
$$

$$
= \mathcal{O}\left(\frac{1}{t}\right).
$$

Furthermore, for every $T \ge 1$, let $\tilde{t}$ be chosen uniformly at random in $\{0,\dots,T-1\}$. Then

$$
\mathbb{E}\left[ 2\sum_{i=1}^n \mathcal{D}_{f_i}^{\tilde{t}} + 4n\mathcal{D}_{f_s}^{\tilde{t}} + 2n\mathcal{D}_g^{\tilde{t}} + \frac{p_{\tilde{t}+1}\chi}{\gamma}\big\|D\hat{\mathbf{x}}^{\tilde{t}}\big\|_{\boldsymbol{\mathcal{U}}}^2 + \frac{p_{\tilde{t}+1}\chi_y}{\gamma}\big\|D_y\hat{\mathbf{x}}^{\tilde{t}}\big\|_{\boldsymbol{\mathcal{U}}_y}^2 \right]
$$

$$
= \frac{1}{T}\sum_{t=0}^{T-1}\mathbb{E}\left[ 2\sum_{i=1}^n \mathcal{D}_{f_i}^t + 4n\mathcal{D}_{f_s}^t + 2n\mathcal{D}_g^t + \frac{p_{t+1}\chi}{\gamma}\big\|D\hat{\mathbf{x}}^t\big\|_{\boldsymbol{\mathcal{U}}}^2 + \frac{p_{t+1}\chi_y}{\gamma}\big\|D_y\hat{\mathbf{x}}^t\big\|_{\boldsymbol{\mathcal{U}}_y}^2 \right]
$$

$$
\le \frac{1}{T}\left( \frac{1}{\gamma}\big\|\mathbf{x}^0-\mathbf{x}^\star\big\|_{\boldsymbol{\mathcal{X}}}^2 + \frac{\gamma a}{\eta b}\big\|\mathbf{u}^0-\mathbf{u}^\star\big\|_{\boldsymbol{\mathcal{U}}}^2 + \frac{\gamma a}{\eta b}\big\|u_y^0-u_y^\star\big\|_{\boldsymbol{\mathcal{U}}_y}^2 \right).
$$

Thus, given $\epsilon > 0$, by choosing

$$
T \ge \frac{1}{2\epsilon}\left( \frac{1}{\gamma}\big\|\mathbf{x}^0-\mathbf{x}^\star\big\|_{\boldsymbol{\mathcal{X}}}^2 + \frac{\gamma a}{\eta b}\big\|\mathbf{u}^0-\mathbf{u}^\star\big\|_{\boldsymbol{\mathcal{U}}}^2 + \frac{\gamma a}{\eta b}\big\|u_y^0-u_y^\star\big\|_{\boldsymbol{\mathcal{U}}_y}^2 \right),
$$

we have

$$
\mathbb{E}\left[ 2\sum_{i=1}^n \mathcal{D}_{f_i}^{\tilde{t}} + 4n\mathcal{D}_{f_s}^{\tilde{t}} + 2n\mathcal{D}_g^{\tilde{t}} + \frac{p_{\tilde{t}+1}\chi}{\gamma}\big\|D\hat{\mathbf{x}}^{\tilde{t}}\big\|_{\boldsymbol{\mathcal{U}}}^2 + \frac{p_{\tilde{t}+1}\chi_y}{\gamma}\big\|D_y\hat{\mathbf{x}}^{\tilde{t}}\big\|_{\boldsymbol{\mathcal{U}}_y}^2 \right] \le 2\epsilon.
$$

In particular,

$$\mathbb{E}\left[\sum_{i=1}^{n}\mathcal{D}_{f_i}^{\tilde{t}} + 2n\mathcal{D}_{f_s}^{\tilde{t}} + n\mathcal{D}_g^{\tilde{t}}\right] \le \epsilon. \tag{21}$$

Also, since $(p_t)_t$ is decreasing,

$$\frac{p_T\chi}{\gamma}\left\|D\hat{\mathbf{x}}^{\tilde{t}}\right\|_{\boldsymbol{\mathcal{U}}}^2 + \frac{p_T\chi_y}{\gamma}\left\|D_y\hat{\mathbf{x}}^{\tilde{t}}\right\|_{\boldsymbol{\mathcal{U}}_y}^2 \le \frac{p_{\tilde{t}+1}\chi}{\gamma}\left\|D\hat{\mathbf{x}}^{\tilde{t}}\right\|_{\boldsymbol{\mathcal{U}}}^2 + \frac{p_{\tilde{t}+1}\chi_y}{\gamma}\left\|D_y\hat{\mathbf{x}}^{\tilde{t}}\right\|_{\boldsymbol{\mathcal{U}}_y}^2,$$

so that

$$\mathbb{E}\left[\frac{\chi}{\gamma}\left\|D\hat{\mathbf{x}}^{\tilde{t}}\right\|_{\boldsymbol{\mathcal{U}}}^2 + \frac{\chi_y}{\gamma}\left\|D_y\hat{\mathbf{x}}^{\tilde{t}}\right\|_{\boldsymbol{\mathcal{U}}_y}^2\right] \le \frac{2}{\epsilon p_T} = 2\epsilon\sqrt{\frac{a+T}{b}},$$

and if $T = \Theta\left(\frac{1}{2\epsilon}\left(\frac{1}{\gamma}\left\|\mathbf{x}^0 - \mathbf{x}^\star\right\|_{\boldsymbol{\mathcal{X}}}^2 + \frac{\gamma a}{\eta b}\left\|\mathbf{u}^0 - \mathbf{u}^\star\right\|_{\boldsymbol{\mathcal{U}}}^2 + \frac{\gamma a}{\eta b}\left\|u_y^0 - u_y^\star\right\|_{\boldsymbol{\mathcal{U}}_y}^2\right)\right)$,

$$\mathbb{E}\left[\frac{\chi}{\gamma}\left\|D\hat{\mathbf{x}}^{\tilde{t}}\right\|_{\boldsymbol{\mathcal{U}}}^2 + \frac{\chi_y}{\gamma}\left\|D_y\hat{\mathbf{x}}^{\tilde{t}}\right\|_{\boldsymbol{\mathcal{U}}_y}^2\right] = \mathcal{O}\left(\sqrt{\epsilon}\right).$$

Finally, the expectation of the number of communication rounds over the first $T \ge 1$ iterations is

$$\sum_{t=1}^{T} p_t = \Theta(\sqrt{T}),$$

so that (21) is achieved with $\Theta(\sqrt{T}) = \Theta\left(\frac{1}{\sqrt{\epsilon}}\right)$ communication rounds.