# OpenReview forum: "BiCoLoR: Communication-Efficient Optimization with Bidirectional Compression and Local Training"
_ICLR.cc/2026/Conference — Submitted to ICLR 2026_

### Official Review · Reviewer_9CAD · 2025-11-01

**Soundness:** 4
**Presentation:** 2
**Contribution:** 2
**Rating:** 4
**Confidence:** 3

**Summary:**

This paper presents BiColor, a new algorithm for distributed optimization that combines local training and bi-directional compression. BiColor achieves state-of-the-art total communication complexity and outperforms other methods in communication cost experiments.

**Strengths:**

- The inclusion of downstream compression is an important contribution in distributed optimization.
- The BiCoLoR method is novel and it achieves SOTA total communication complexity.

**Weaknesses:**

-	The objective function form is quite specific, and it is not clear how generally applicable it is.  The paper cites empirical risk minimization as an application of this formulation, but it does not give any other applications. Further, the convexity assumptions limit its applicability in Federated Learning settings, despite this being the target setting.

- The presentation is challenging for someone not familiar with this specific line of work. The paper devotes significant space to compression (Section 1.1), when it is not clear all of this detail is needed. The space would be better used for the experiments that were deferred to the appendix.
- Also w.r.t. presentation, even though there is a lot of detail about the related work in Section 2, I am still having trouble putting the proposed method in context. In particular, it seems the total communication complexity of BiCoLor and 2Direction are the same, and so the theoretical benefits of BiCoLor are not clear.
- I had a similar presentation issue with Section 4. There is a great deal of space devoted to the theorem and corollaries, and the discussion of the total communication complexity is deferred to the appendix. This is a strange choice given this result is the main technical contribution.

**Questions:**

- Do all of the related works in Table 1 address the same problem formulation? If not, it would be good to include some details in the table about the types of problems that the methods can address.

---

### Official Review · Reviewer_FN6z · 2025-11-01

**Soundness:** 2
**Presentation:** 1
**Contribution:** 1
**Rating:** 2
**Confidence:** 3

**Summary:**

The paper proposes BiCoLoR, which merges Local Training (LT) with bidirectional compression (BiCC) under arbitrary unbiased compressors on both uplink and downlink. The key design decouples uplink/downlink errors so they add instead of multiply, and triggers communication stochastically.

**Strengths:**

+ The algorithmic plumbing (server sends its own compressed difference; clients don’t receive the aggregated uplink average) keeps uplink/downlink stochasticity independent, avoiding the typical multiplicative variance blow-up and enabling sharper complexity.
+ Table-style comparisons in text relate BiCoLoR to MURANA, MCM, EF21-P+DIANA, and 2Direction; BiCoLoR achieves the same asymptotic TotalCom without occasional full-precision sends.
+ Theorem 4.1 specifies step sizes (ρ,η) delivers linear convergence, and derives communication/bit complexities.

**Weaknesses:**

- There is no conclusion part in this paper.

- Empirical scope is thin. The only shown experiments (logistic regression on real-sim) are informative but narrow; there’s no large-scale non-IID, partial participation, or heterogeneous latency study, which matters for BiCC realism.

- Systems realism left implicit. The TotalCom model counts bits but omits control-plane costs (index/header overhead for sparsification/quantization, compressor seed sync, server broadcast fan-out), queueing, and stragglers—important for downlink-heavy regimes (α≈1).

**Questions:**

- Can you follow the ICLR format with conclusion parts included?
- In highly heterogeneous settings, how sensitive are the gains to 𝜔?

	​

---

### Official Review · Reviewer_5ByQ · 2025-11-02

**Soundness:** 2
**Presentation:** 1
**Contribution:** 2
**Rating:** 4
**Confidence:** 3

**Summary:**

The paper addresses computation and communication costs by coupling local training with bidirectional (uplink/downlink) compression. The authors establish convergence guarantees, analyze communication complexity, and demonstrate empirical gains over existing algorithms.

**Strengths:**

This paper studies communication efficiency, a key challenge in distributed optimization, by adopting a realistic setting that reduces both uplink and downlink bandwidth. The authors provide convergence guarantees, analyze communication complexity, and validate the approach empirically.  Without requiring transmitting full vectors with small probability, the algorithm achieves a similar bound on total communication complexity. The experimental results show the better performance compared to existing algorithms, in terms of communication costs.

**Weaknesses:**

1. The writing needs to be improved: There is no conclusion section and exist several grammar issues. Many abbreviations that hinder readability. Some notations is used with definition or with multiple meanings (like \phi in Line 119-124).
2. The algorithm is too complicated, with many hyperparameters and moving parts; a schematic or simplified pseudocode would improve accessibility. The algorithm appears to build on LoCoDL and bidirectional compression ideas. It would be better if the authors could clarify the challenges of extending LoCoDL to the two direction compression setting and the inherent challenges of analysis.
3. The experimental section is relegated to the appendix, and key details are missing. For example, what is the dataset size m? What is the probability that 2Direction communicates using full vectors?

**Questions:**

1. Does the improvement of the variances from the product to the sum come from the uplink and downlink independence assumption?
2. What are the main challenges in avoiding occasional full, non-compressed vector transmissions (used with small probability in prior work)?
3. Does the proposed algorithm outperform the other algorithms for smaller \alpha?

---

### Official Review · Reviewer_czrG · 2025-11-08

**Soundness:** 4
**Presentation:** 3
**Contribution:** 3
**Rating:** 6
**Confidence:** 3

**Summary:**

This work studies federated learning (FL) where both uplink (clients to server) and downlink (server to client) costs can be expensive. More commonly in FL, only the uplink cost is considered and downlink communication costs are generally ignored.

This work studies a setting where the total communication cost (TotalCom) is a sum of UpCom + $\alpha \cdot$ DownCom for a parameter $\alpha \in [0, 1]$. If $\alpha = 0$, then the server to client communication cost is considered zero, and if $\alpha = 1$, then downlink and uplink costs are symmetric and equally expensive.

The authors study how local training (LT), that is, each client making a few local updates before sending signals to the server, and bidirectional compression (BiCC), that is, messages sent by the clients to the server and back from the server to the clients being compressed using lossy compression, can reduce the number of bits communicated.

One key observation is that naive bidirectional compression tends to multiply the errors because the server aggregates lossy compressed messages from the clients to make updates and then compresses the updates again, so the errors pile up.

The authors introduce an algorithm called BiCoLoR which addresses bidirectional communication by making the uplink and downlink compressions independent and broadcasting differences of updates, where the difference is with respect to a shared parameter $y$ that is maintained the same on all client and server machines.

To achieve this effect, instead of considering the traditional FL objective of minimizing $\sum_{i=1}^n f_i(x)$, the authors consider $\sum_{i=1}^n f_i(x) + f_s(x) + g(x)$. Here $f_s(x)$ acts like an objective for the server so that it can calculate gradients for it, so in a way the server acts like the $(n+1)$th client. The presence of $g(x)$ lets the clients and server share parameters and reference points, allowing transmission of compressed differences.

In BiCoLoR, the clients compute local updates using a dual variable and $f_i$, and the shared update $\hat{y}$ using the gradient of $g$. With probability $p_t$, they send the compressed difference of the local update and $\hat{y}$ to the server, and the server broadcasts compressed updates back. The paper uses a combination of rand-$K$ compressors, which randomly select and scale $K$ out of $d$ gradient coordinates, with a general class of unbiased compressors with bounded relative variance.

In the strongly convex case, BiCoLoR achieves linear convergence in the number of total rounds; the TotalCom is bounded by $\mathcal{O}\\left((K + \alpha K_s)\left(\frac{d \sqrt{\kappa}}{\sqrt{K K_s}} + \frac{d^2}{K K_s} \log \frac{1}{\epsilon}\right)\right)$.

When $\alpha$ is close to $0$, this becomes $\mathcal{O}(\sqrt{d \kappa} + d)$, and it is $\mathcal{O}(d\sqrt{\kappa})$ when $\alpha$ is not too small.

The authors also prove a result for the general convex case and show that using a decreasing $p_t$ can reduce total communication rounds to $\mathcal{O}\\left(\sqrt{\frac{L}{\epsilon}}\right)$.

The paper also includes experiments showing BiCoLoR achieves better convergence in terms of bits compared to baselines.

\section{What I like}
The problem considered in the paper is novel and highly relevant to the conference.

**Strengths:**

The problem considered in the paper is novel and highly relevant to the conference.

The paper considers and decouples uplink and downlink costs in distributed optimization. This is done by considering an extra shared parameter $y$ and only communicating differences to this common $y$, resulting in the error variance being added instead of multiplied.

The algorithm is versatile and gets accelerated guarantees in both strongly convex and general convex settings. The paper also discusses why improving the $d$ dependence is difficult.

The empirical results are extensive, compare shared versus independent rand-$K$, and show shared $k$ works well in practice.

Overall the paper is well written and motivated.

**Weaknesses:**

In the case where $\alpha$ is not too small, the TotalCom is the same as standard accelerated gradient descent (AGD), i.e., $\tilde{\mathcal{O}}(d \sqrt{\kappa})$. Even though AGD sends full gradients, its faster convergence rate than BiCoLoR lets it achieve the same TotalCom. So the results of BiCoLoR show that compression can get similar TotalCom, but not better. The authors do discuss why getting better convergence in terms of $d$ would be harder.

This is even without considering the bits required to communicate which $K$ coordinates are sent in the updates, and to achieve these results the optimization problem needs to ensure that there is a shared loss $g$ between the clients and the server.

In the case where $\alpha$ is small, the results are not optimal, and another algorithm, ADIANA, in fact has better convergence. The authors discuss and show how BiCoLoR can be more competitive empirically.

In the empirical analysis, the authors do not compare performance against ADIANA or AGD.

**Questions:**

1. How do the results change if we consider the additional cost of transmitting the bits for the positions for rand-$K$?
2. If alpha is small, ADIANA has better convergence, and when alpha is bigger, AGD has similar TotalCom, so when do the authors actually recommend BiCoLor to be more useful or superior than existing approaches?
3. How does BiCoLoR perform with respect to AGD or ADIANA empirically?
4. How realistic is to assume that the server and clients will have a shared function to optimize?
5. How does BiCoLoR perform with respect to the baselines in the partial participation case?

---

### Author Response · Authors · 2025-11-27

We thank the reviewers for their time and efforts. Given the rather negative scores, in particular one reviewer giving a score of 2 with off-topic comments, we are not going to waste our time in the rebuttal process. Thank you for your understanding.

---

### Meta-Review · Area_Chair_na6d · 2026-01-04

**Summary:**

Some reviewers like the problem studied and the algorithm design in this work, however, they all raised concerns on presentation/writing, theoretical results, and experiments. I agree with the authors that some reviews are negative and relatively short, it was unfortunate that the authors didn't provide a rebuttal to address them, so it was very hard to justify if the concerns can be addressed. Given the reviews, and the unaddressed concerns, after carefully reading the submission I recommend reject.

**Reviewer Concerns:**

Reviewers have concerns on presentation/writing, theoretical results, and experiments, which remain unaddressed due to the absence of an author rebuttal.

**Reviewer Scores:**

Reviewer scores won't change since there is no author rebuttal.

---

### Decision · Program_Chairs · 2026-01-26

Reject